# Improving the Performance of Surface Flow Generated by Bubble Plumes

Hassan Abdulmouti 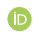

Department of Mechanical Engineering Division, Sharjah Men's College, Higher Colleges of Technology, Sharjah P.O. Box 7946, United Arab Emirates; habdulmouti@hct.ac.ae or hassanabujihad@hotmail.com; Tel.: +971-22066557

**Abstract:** Gas–liquid two-phase flow is widely used in many engineering fields, and bubble dynamics is of vital importance in optimizing the engineering design and operating parameters of various adsorptive bubble systems. The characteristics of gas–liquid two-phase (e.g., bubble size, shape, velocity, and trajectory) remain of interest because they give insight into the dynamics of the system. Bubble plumes are a transport phenomenon caused by the buoyancy of bubbles and are capable of generating large-scale convection. The surface flow generated by bubble plumes has been proposed to collect surface-floating substances (in particular, oil layers formed during large oil spills) to protect marine systems, rivers, and lakes. Furthermore, the surface flows generated by bubble plumes are important in various types of reactors, engineering processes, and industrial processes involving a free surface. The bubble parameters play an important role in generating the surface flow and eventually improving the flow performance. This paper studies the effects of temperature on bubble parameters and bubble motion to better understand the relationship between the various bubble parameters that control bubble motion and how they impact the formation of surface flow, with the ultimate goal of improving the efficiency of the generation of surface flow (i.e., rapidly generate a strong, high, and wide surface flow over the bubble-generation system), and to control the parameters of the surface flow, such as thickness, width, and velocity. Such flow depends on the gas flow rate, bubble size (mean bubble diameter), void fraction, bubble velocity, the distance between bubble generator and free surface (i.e., water height), and water temperature. The experiments were carried out to measure bubble parameters in a water column using the image visualization technique to determine their inter-relationships and improve the characteristics of surface flow. The data were obtained by processing visualized images of bubble flow structure for the different sections of the bubble regions, and the results confirm that temperature, bubble size, and gas flow rate significantly affect the flow structure and bubble parameters.

**Keywords:** multiphase flow; fluid; bubble plume; bubble; surface flow; thermal; bubble flow

## 1. Introduction

In recent years, multifluid systems, including the gas injection technique (which is the most popular model in the field of bubble dynamics), have been widely used and play an important role in many natural and industrial processes and many engineering fields such as materials; combustion; petroleum refining; chemical, mechanical, and environmental engineering; cleaning, heat and mass transfer promoting, and high-pressure evaporators. They also have been used to improve chemical reactions, waste treatment, gas mixing, resolution, and other engineering processes [1–5]. Furthermore, because of their simple construction and ease of operation, bubble columns are widely used in the petrochemical, pharmaceutical, and metallurgical industries as multiphase reactors and contactors. Moreover, the gas-liquid two-phase flow has seen wide use in many areas, such as biological fermentation, polymer polymerization, industrial wastewater treatment, and environmental protection [6–10]. Although the gas injection model has been used

with varying degrees of success to describe bubble plumes, more information on this subject is needed because generating surface flow can still be made more efficient. The rising of a buoyancy-driven bubble in a liquid is a typical process in multifluid systems, so a sound understanding of the fundamentals of rising bubbles is crucial in a variety of practical applications, ranging from the rise of steam in boiler tubes to gas bubbles in oil wells. Because of the strong nonlinearity accompanied by large bubble deformations, it is difficult to study the mechanisms behind bubble behavior solely through theoretical methods [11–18].

The functionality of bubble column dependence on the velocity of an air bubble has been determined experimentally by numerous investigators. When rising through an infinite stagnant liquid, the single bubble's terminal velocity is of fundamental importance in two-phase flow [19], as well as in other types of flow. The knowledge of bubble properties, including bubble velocity, bubble size, gas holdup, and specific interfacial area, is of prime importance for the proper design and operation of bubble columns [20,21]. An understanding of bubble–fluid interactions is important in a broad range of natural, engineering, and medical settings. Air–sea gas transfer, bubble column reactors, oil and/or natural gas transport, boiling heat transfer, ship hydrodynamics, ink-jet printing, and medical ultrasound imaging are just a few examples where the dynamics of bubbles play a role [22–25].

The movement of bubbles is a basic subject in gas–liquid two-phase flow research because bubbles in water play an important role in solving problems in a wide range of experiments and projects [26–28]. The state and motion of bubbles are closed within the operating conditions, the nature of the liquid, and the form of ventilation. Implementing a bubble state by using visual methods has helped the development of the chemical industry and related fields [29–31]. The motion of bubbles is very complex. The upward path and change in the direction of a bubble are known to be strongly related to bubble shape [32]. The motion of spherical bubbles is usually rectilinear. Once the bubble deforms into an oblate ellipsoid, instability sets in and results in a spiral or zigzag trajectory. Both bubble shape and bubble velocity of oblate ellipsoidal bubbles exhibit chaotic features [33,34]. In turn, the fluctuation of bubble shape is likely to cause oscillations in the drag force, leading to the chaotic fluctuation of bubble velocity in the streamwise direction. Therefore, despite periodic macroscopic motion, bubbles exhibit highly chaotic fluctuations in both the lateral and axial components along the zigzag path of a bubble ascent. At the same time, bubble orientation changes in such a way that the trajectory of the bubble plume tends to be perpendicular to the direction of instantaneous motion [35–37]. From gas-disengagement experiments, it was inferred that both large and small bubbles exist in churn-turbulent flow [38–40].

Large bubbles rise fast through the column, whereas small bubbles display a longer resident-time distribution. Beyond a certain transition gas velocity, the small bubble holdup remains constant, whereas the large bubble holdup continues to increase with gas velocity [41–43]. Bubbles in motion are generally classified as spherical, oblate ellipsoidal, or ellipsoidal cap, etc. In gas–liquid upward flow, bubbles move faster than the surrounding liquid (due to buoyancy), and large bubbles have greater upward acceleration than small bubbles. The actual bubble shape depends on the relative magnitudes of the forces acting on the bubble, such as surface tension and inertial forces [44–47].

The behavior and characteristic of the surface flow induced by a bubble plume was determined by H. Abdulmouti. This behavior depends on the gas flow rate, bubble size, and internal two-phase flow structure of the bubbly flow. The author carried out two-dimensional flow analysis based on the Eulerian-Lagrangian model and particle tracking velocimetry measurement to elucidate the surface flow generation process. The author predicted the surface velocity profile by the Eulerian-Lagrangian model and calculated the maximum velocity of the surface flow as a function of bubble size. The author performed a numerical analysis and experimental study of a surface flow generation process by buoyant bubbles within the two-dimensional and three-dimensional flow analysis supported by the

Eulerian-Lagrangian model [48–50]. The entire flow around the bubble flow was clarified experimentally by PIV measurement by H. Abdulmouti et al., who explained the mechanism of generating the surface flow under the free surface and the wave damping effect of the surface flow [51]. H. Abdulmouti elucidated during a review paper the characteristics, structures, behaviors, and flow patterns of bubbly two-phase flow [52–55]. H. Abdulmouti conducted experimental measurements of bubble convection models in two-phase stratified liquids, where the author described the mechanism, the characteristics, and therefore the multi-dimensional motion of the flow around the plume in two stratified layers of water and oil by using particle imaging velocimetry (PIV) measurement and pathline measurements. Furthermore, the author measured the liquid flow pattern of bubble-induced convection by using thermo-sensitive liquid tracer particles [56,57]. Y. Murai et al. and Hassan demonstrated the mechanism of the flow within the vicinity of the free surface induced by a bubble plume by using numerical simulation [58,59]. H. Abdulmouti administered an experimental measurement for surface flow characteristics generated by a bubble plume, where the author evaluated the speed of the surface flow and its characteristics; the author also interpreted how the surface flow is generated by the bubble plume [60]. H. Abdulmouti elaborated on the 2D numerical simulation of surface flow velocity and therefore the internal flow structure generated by bubbles [61]. Hassan illustrated during a reviewed paper the characteristics and parameters that cover the bubbly two-phase flow [62]. H. Abdulmouti and Hassan studied a visualization and image measurement for the flow structures induced by a bubbly plume [63,64]. Hassan and H. Abdulmouti et al. reported the applicability of a replacement bubble curtain sort of oil fence that decreases the intensity of wave motion and restrains the waves passing over an oil fence body [65]. A few publications (including those cited in the discussion above) describe how temperature is related to and affects the bubble parameters and bubble motion. However, these papers do not explain the relationship between the bubble parameters that control bubble motion, nor do they describe how to increase the efficiency with which surface flow is generated. The image processing used in this paper has advancement in the image acquisition technique of clarity and higher accuracy in identifying each bubble to find the bubble center, the bubble size, and the bubble velocity. These calculations use the time average of 250 consecutive frames from the image processing software (corresponding to 5 s of flow visualization). The average bubble diameter, bubble velocity, void fraction, and standard deviation are calculated by measuring over 20,000 bubbles in the local video images inside the bubble plume. These images were produced by taking local pictures of different regions. The experiments for measuring these parameters were done in three regions of the bubble plume: the first region was over the injector region of the bubble generator, the second was in the middle region of the bubble plume, and the third was just under the free surface. The above-mentioned kinds of literature hardly show an accurate technique similar to our technique in calculating these parameters. More details on the advancement in the image acquisition technique will be explained in a later section. The structure of this manuscript is explained as follows: the experimental apparatus and method of conducting the experiments are explained in detail in part 2, including the conditions of performing the experiments. In part 3, the methods of calculating the bubble parameters are explained in detail. Then the interpretation and discussions of the results of this work are demonstrated in part 4, while the concluding marks are listed in part 5. The above extensive literature review was performed to explain the physics and complexity of bubble motions in liquid columns. By using the observations from our experimental data, we could explain how bubble size and velocity can impact the formation of surface flow, which can potentially be used in real-world applications.

## 2. Experimental Apparatus, Method, and Conditions

The experimental apparatus for investigating how temperature affects the bubble parameters and bubble motion is shown in Figure 1. The interior of the original tank was 750 mm long, 500 mm high, and 200 mm wide and made of transparent acrylic resin

(Figure 1a). To experiment with different geometries and cover different regions of parameter space for the bubbles (gas flow rate, water temperature, water height, bubble velocity, bubble size, and void fraction), the tank height was extended to 1200 mm (Figure 1b). This allowed the gas injection model to be validated because it led to more accurate experimental results. The experimental conditions are given in Table 1. The bubble generator was installed at the bottom center of the tank. The bubble generator was 300 mm long, 20 mm high, and 20 mm wide and made of transparent acrylic resin with 40 holes in the liner to avoid overlapping and/or coalescing bubbles and to assure accurate two-dimensional measurements. Each hole was 0.8 mm in diameter and separated by 7 mm from neighboring holes. The gas flow rate was precisely controlled by a pressure regulator and a flowmeter. Four heaters including thermometers with thermostats and sensors were used to read and control the water temperature in order to allow a range from 20 to 50 °C to maintain the water temperatures. The heaters including thermometers with thermostats and sensors were located and distributed uniformly and equally in the right and lift sides of the upper and lower part of the tank. A lighting setup (direct lighting method) with a black background sheet and two 1000 W light-emitting diodes were used to clearly visualize and capture images of the flow for the measurements. The visualized flows were recorded by a digital video camera (Panasonic HC-MDH2) at 50 frames per second. The digital images were preprocessed by video-to-JPEG converter software and Adobe After Effects CS6 image processing software. The preprocessing entailed sharpening, binarizing, smoothing the images, and labeling bubbles. The code image processing software was used to enable identification of each bubble after preprocessing the images. The images were converted to the computer to treat them by our image acquisition technique and software. In the image processing of our code, any noises or unknown objects in the images were first eliminated if available, and then the images were smoothed to make them clearer and readier for bubble identification; after that, the boundary of each bubble was identified and binarized. The bubbles that had unclear boundaries because of light reflection, including distortion of the observed bubble size due to refraction, or other reasons were sharpened then binarized. These popular kinds of preprocessing are necessary for many kinds of software or codes and usually do not affect the sensitivity of the results of the parameter calculations. This kind of preprocessing can be shown in many kinds of literatures, such as [29,35,48,51,56,58–61]. In the recorded images, the bubble plume is located in the middle of the image, and our observation showed that the bubbles have an almost two-dimensional motion in the *x-y* plane, especially for low and medium gas flow rates, except for a tiny degree of three-dimensional fluctuations due to turbulence of the flow with high gas flow rates. The bubbles were considered to be two dimensions while indeed they were three dimensions, but this consideration was accepted in our calculation, as it did not affect the calculations, and it was considered in the measurement uncertainty. The velocity of the flow is almost two-dimensional especially because the tank width is small comparing to other tank dimensions and there are no perpendicular components to the front and back walls in the flow field. The uncertainty of measurements can be from various sources such as the reference measurement device used to make the measurement, environmental conditions, the operator making the measurements, the procedure, the calculations of the image processing which is the most important factor, and many other sources. In our experiments, we made our best effort to minimize these factors, especially given that the experiments were repeated many times. The image processing used in this paper has advancement in the image acquisition technique of clarity and higher accuracy in identifying each bubble to find the bubble center, the bubble size, and the bubble velocity. Our calculations used the time average of 250 consecutive frames from the image processing software (corresponding to 5 s of flow visualization). The average bubble diameter, bubble velocity, void fraction, and standard deviation were calculated by measuring over 20,000 bubbles in the local video images inside the bubble plume. The "root sum of the squares" methods and the standard deviations methods were applied to obtain higher accuracy (square the value of each uncertainty source). The combined standard uncertainty was calculated by squaring

the value of each uncertainty component and adding together all the results and then calculating the square root of the result by finding the sum of squares. The result shows the combined standard uncertainty that we obtained. The dimensions of these measurements are the distance of the bubble movements between one frame and another, which affects the bubble velocity and void fraction, the two dimensions of the area for calculating the void fraction $A$, and the two dimensions of the bubble size that affects the bubble diameters, as will be explained in later sections.

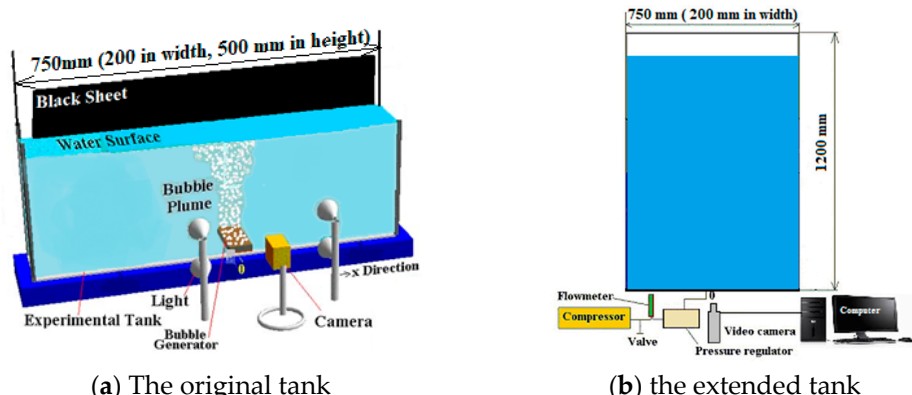

(**a**) The original tank          (**b**) the extended tank

**Figure 1.** Schematic diagram of experimental apparatus.

**Table 1.** Experimental conditions.

| Parameter | Value |
|---|---|
| Density of water | $\rho = 1000 \text{ kg/m}^3$ |
| Kinematic viscosity of water | $\upsilon = 0.547 \times 10^{-6} \text{ to } 1 \times 10^{-6} \text{ m}^2/\text{s}$ |
| Initial water depth | $H = 0.1 \text{ to } 1.2 \text{ m}$ |
| Atmospheric pressure | 101 kPa |
| Ambient temperature | 22–25 °C |
| Water temperature | 20–50 °C |
| The density of the gas (air) | 1.25 kg/m$^3$ |
| Maximum gas flow rate | $Q = 50 \times 10^{-5} \text{ m}^3/\text{s}$ |

## 3. Calculation of Bubble Parameters

### 3.1. The Mean Bubble Diameter

The mean bubble diameter that represented the bubble size in our study was calculated for water depths from 100 to 1200 mm (corresponding to the change in height of the free surface in the tank) and three ranges of gas volume flow rates ($Q_1$ to $Q_3$). These calculations used the time average of 250 consecutive frames from the image processing software (corresponding to 5 s of flow visualization). The average bubble diameter and standard deviation were calculated by measuring over 20,000 bubbles in the local video images inside the bubble plume. These images were produced by taking local pictures of different regions. The experiments for measuring bubble size were done in three regions of the bubble plume: the first region was over the injector region of the bubble generator, the second was in the middle region of the bubble plume, and the third was just under the free surface. Figure 2 shows samples of bubble images for different bubble-generating conditions in these regions. The first image (left side of Figure 2) shows the bubbles for a gas flow rate $Q_1 = 1.5 \times 10^{-5} \text{ m}^3/\text{s}$, a water height H = 400 mm, and a temperature T = 30 °C. The middle image of Figure 2 shows the bubbles for a gas flow rate $Q_2 = 28 \times 10^{-5} \text{ m}^3/\text{s}$, a water height H = 400 mm, and a temperature T = 30 °C; and the third image (right side of Figure 2) shows the bubbles for a gas flow rate $Q_3 = 50 \times 10^{-5} \text{ m}^3/\text{s}$, a water

height H = 400 mm, and a temperature T = 30 °C. The bubble diameter was defined by the equivalent bubble diameter using ellipsoidal approximations for the bubble shape of each bubble and in each image, which was obtained by using the image possessing software after binarizing the images. The equivalent bubble diameter was estimated from the vertical and horizontal lengths of each bubble in each image. The images were obtained by using the JPEG image-converter software and Adobe After Effects CS6 image processing software after binarizing the images. The minimum bubble size in the experiments was more than 2 mm. This kind of image processing for a bubbly two-phase flow is suitable for this kind of medium and large-scale flow system, it gives a reasonable accuracy, especially when the number of bubbles reaches several thousand. Thus, averaged bubble diameter is necessary to carry out the present accuracy calculation. However, other techniques such as the two-fluid model and normal image processing, which are the most popular local averaged models, do not give high predictability owing to the lack of spatial resolution, especially for the calculation of microbubble size. The microbubble cannot be measured with acceptable accuracy in the coded software in the two-phase region, as it cannot be extracted from this image clearly due to the limitation in the pixel resolution. Hence, in our image acquisition technique, the drawback of the code is that it has a limitation of the calculation where it cannot calculate small bubble or microbubble sizes that have diameters of less than 1 mm. In our experiment cases, more than 98% of the bubbles had sizes of more than 2 mm. Although a few limited numbers of bubbles had sizes smaller than 2 mm, those bubbles were eliminated to obtain higher accuracy. The measurement uncertainty depends on the pixel resolution; for the bubble diameter, it was estimated to be 0.010–0.015 mm, which reflects the accuracy of the experimental results.

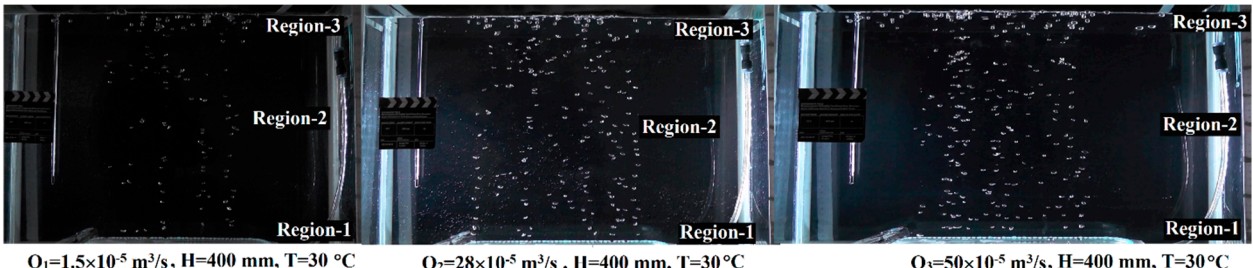

**Figure 2.** Typical images of bubbles.

Figure 3 shows a sample of the bubble diameter as a function of water height in the tank for water temperatures of 20 °C. Where in this figure, the horizontal axis represents the water height (mm), the vertical axis represents the bubble diameter (mm). The upper set of the three lines of the figure illustrates the relationship between the water height and the bubble diameter for $Q_3$, while the middle set is for $Q_2$, and the lower one is for $Q_1$. The green lines indicate the upper region (U) or upper level of the bubble plume inside the tank, while the red lines are for the middle level or region (M), and the blue lines are for the lower region or level (L). The lower level is just above the bubble generator (over the injector region of the bubble generator), the middle level is halfway between the bubble generator and the free surface, and the upper level is just under the free surface.

Tables 2–5 show the values of the bubble diameter as a function of water height in the tank for water temperatures of 20, 30, 40, and 50 °C, respectively. The data were collected for three gas volume flow rates ($Q_1 = 1.5 \times 10^{-5}$ m$^3$/s, $Q_2 = 28 \times 10^{-5}$ m$^3$/s, $Q_3 = 50 \times 10^{-5}$ m$^3$/s). The bubble diameter was calculated at the three heights in the bubble plume: lower (L), middle (M), and upper (U). The first column of the tables shows the water height (mm), and the second column shows the water temperature, while the third set of columns (4 columns) elucidates the values of the bubble diameter for the three levels (lower, middle and upper regions of the tank) for the gas volume flow rate $Q_1 = 1.5 \times 10^{-5}$ (m$^3$/s). The fourth column of this set states the range of uncertainties. The

fourth and the fifth sets are the same for gas volume flow rate $Q_2 = 28 \times 10^{-5}$ (m$^3$/s) and $Q_3 = 50 \times 10^{-5}$ (m$^3$/s), respectively.

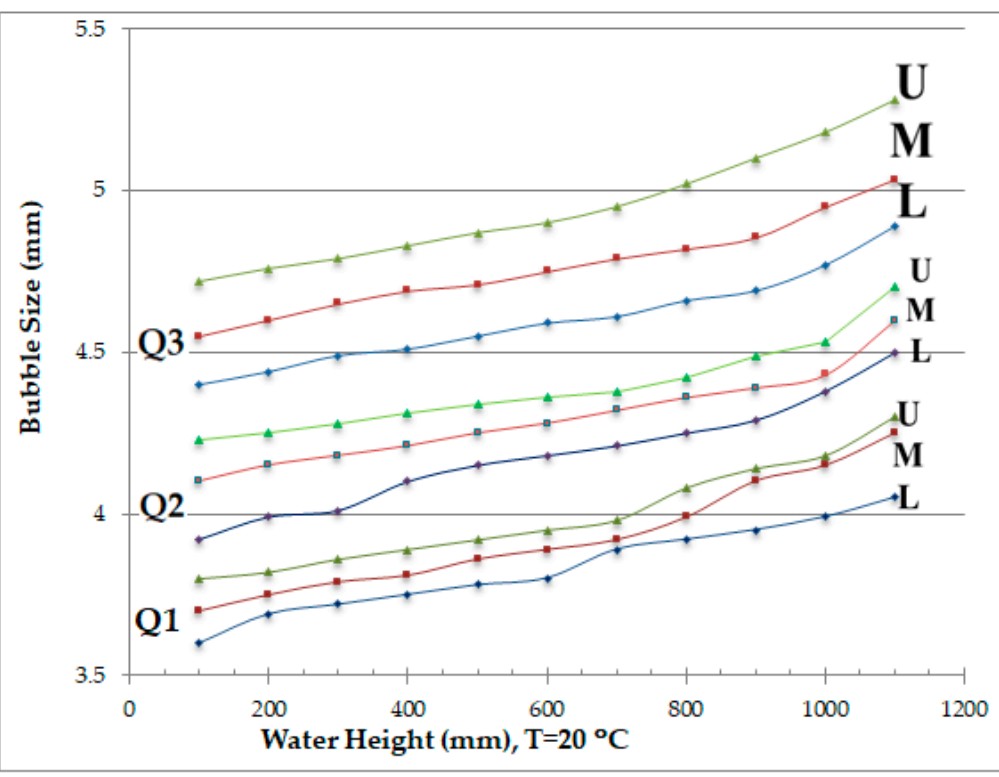

**Figure 3.** A sample of bubble size as a function of water height for $T = 20\,°C$.

The above results showed clearly that the bubble size increased with water height in the tank. These results were consistent with those of our previous studies [66]. Furthermore, the bubble size increased as the water temperature increased. The results showed clearly that the bubble diameter increased about one mm for each 10 °C increase in water temperature. Beyond that, the higher range of uncertainties occurred in the upper region where the flow started to change its orientation from a horizontal into a vertical direction, forming the surface flow, and especially with the higher gas volume flow rate $Q_3 = 50 \times 10^{-5}$ (m$^3$/s), where the flow started to be turbulent.

### 3.2. The Bubble Velocity

The bubble velocity was calculated for water depths from 100 to 1200 mm (corresponding to the change in height of the free surface in the tank) and three ranges of gas volume flow rates ($Q_1$ to $Q_3$) These calculations used the time average of 250 consecutive frames from the image processing software (corresponding to 5 s of flow visualization). The velocity of each bubble was calculated as the vertical velocity after identifying the position and the bubble center in each image and the position of the bubble center of the same bubble in the next consecutive image; then, the distance between those two positions of the consecutive image was computed to be the movement of the bubble from one frame to another. By knowing the speed of the camera between two consecutive images, the bubble velocity and standard deviation were calculated, and hence the average of all bubbles was computed for over 20,000 bubbles in the local video images inside the bubble plume for each level using image processing. These images were produced by taking local pictures of different regions.

**Table 2.** Bubble diameter (mm) as a function of water height for $T = 20\,°C$ and for gas volume flow rates $Q_1$–$Q_3$.

| Water Height (mm) | Water Temperature (°C) | Gas Volume Flow Rate $Q_1 = 1.5 \times 10^{-5}$ (m$^3$/s) | | | | Gas Volume Flow Rate $Q_2 = 28 \times 10^{-5}$ (m$^3$/s) | | | | Gas Volume Flow Rate $Q_3 = 50 \times 10^{-5}$ (m$^3$/s) | | | |
|---|---|---|---|---|---|---|---|---|---|---|---|---|---|
| | | Lower | Middle | Upper | Range of Uncertainties | Lower | Middle | Upper | Range of Uncertainties | Lower | Middle | Upper | Range of Uncertainties |
| 100 | | 3.600 | 3.700 | 3.800 | 0.01000–0.01010 | 3.920 | 4.100 | 4.230 | 0.01047–0.01051 | 4.400 | 4.580 | 4.720 | 0.01075–0.01079 |
| 200 | | 3.690 | 3.750 | 3.820 | 0.01005–0.01011 | 3.990 | 4.150 | 4.250 | 0.01052–0.01053 | 4.440 | 4.600 | 4.760 | 0.01080–0.01083 |
| 300 | | 3.720 | 3.790 | 3.860 | 0.01014–0.01019 | 4.010 | 4.180 | 4.280 | 0.01055–0.01058 | 4.490 | 4.650 | 4.790 | 0.01084–0.01088 |
| 400 | | 3.750 | 3.810 | 3.890 | 0.01020–0.01023 | 4.100 | 4.210 | 4.310 | 0.01056–0.01058 | 4.510 | 4.690 | 4.830 | 0.01088–0.01093 |
| 500 | | 3.780 | 3.860 | 3.920 | 0.01024–0.01029 | 4.150 | 4.250 | 4.340 | 0.01057–0.01059 | 4.550 | 4.710 | 4.870 | 0.01094–0.01097 |
| 600 | T = 20 | 3.801 | 3.890 | 3.950 | 0.01030–0.01032 | 4.180 | 4.280 | 4.360 | 0.01059–0.01061 | 4.590 | 4.750 | 4.901 | 0.01096–0.01098 |
| 700 | | 3.890 | 3.920 | 3.980 | 0.01033–0.01036 | 4.210 | 4.320 | 4.380 | 0.01062–0.01064 | 4.610 | 4.790 | 4.950 | 0.01097–0.01100 |
| 800 | | 3.920 | 3.990 | 4.080 | 0.01037–0.01039 | 4.250 | 4.360 | 4.420 | 0.01063–0.01065 | 4.660 | 4.820 | 5.020 | 0.01099–0.01105 |
| 900 | | 3.950 | 4.102 | 4.140 | 0.01039–0.01041 | 4.290 | 4.390 | 4.490 | 0.01065–0.01067 | 4.690 | 4.860 | 5.150 | 0.01107–0.01109 |
| 1000 | | 3.990 | 4.150 | 4.180 | 0.01042–0.01044 | 4.380 | 4.430 | 4.530 | 0.01066–0.01068 | 4.720 | 4.900 | 5.200 | 0.01115–0.01122 |
| 1100 | | 4.050 | 4.200 | 4.300 | 0.01045–0.01047 | 4.400 | 4.600 | 4.700 | 0.01069–0.01070 | 4.890 | 5.000 | 5.280 | 0.01128–0.01135 |

**Table 3.** Bubble diameter (mm) as a function of water height for $T = 30\,°C$ and for gas volume flow rates $Q_1$–$Q_3$.

| Water height (mm) | Water Temperature (°C) | Gas Volume Flow Rate $Q_1 = 1.5 \times 10^{-5}$ (m$^3$/s) | | | | Gas Volume Flow Rate $Q_2 = 28 \times 10^{-5}$ (m$^3$/s) | | | | Gas Volume Flow Rate $Q_3 = 50 \times 10^{-5}$ (m$^3$/s) | | | |
|---|---|---|---|---|---|---|---|---|---|---|---|---|---|
| | | Lower | Middle | Upper | Range of Uncertainties | Lower | Middle | Upper | Range of Uncertainties | Lower | Middle | Upper | Range of Uncertainties |
| 100 | | 4.445 | 4.531 | 4.577 | 0.01097–0.01099 | 4.609 | 4.683 | 4.888 | 0.01143–0.01147 | 4.994 | 5.030 | 5.100 | 0.01182–0.01188 |
| 200 | | 4.453 | 4.534 | 4.583 | 0.01105–0.01109 | 4.609 | 4.729 | 4.894 | 0.01147–0.01151 | 5.048 | 5.050 | 5.200 | 0.01186–0.01191 |
| 300 | | 4.467 | 4.557 | 4.600 | 0.01114–0.01116 | 4.630 | 4.743 | 4.967 | 0.01152–0.01155 | 5.099 | 5.180 | 5.300 | 0.01190–0.01195 |
| 400 | | 4.470 | 4.564 | 4.600 | 0.01115–0.01117 | 4.638 | 4.867 | 4.980 | 0.01156–0.01159 | 5.159 | 5.250 | 5.400 | 0.01194–0.01198 |
| 500 | | 4.475 | 4.577 | 4.642 | 0.01116–0.01119 | 4.787 | 4.888 | 4.994 | 0.01160–0.01164 | 5.299 | 5.356 | 5.400 | 0.01197–0.01200 |
| 600 | T = 30 | 4.479 | 4.583 | 4.653 | 0.01119–0.01123 | 4.809 | 4.894 | 5.048 | 0.01163–0.01167 | 5.299 | 5.410 | 5.500 | 0.01199–0.01205 |
| 700 | | 4.480 | 4.600 | 4.700 | 0.01124–0.01127 | 4.849 | 4.967 | 5.099 | 0.01169–0.01172 | 5.306 | 5.400 | 5.600 | 0.01209–0.01215 |
| 800 | | 4.485 | 4.609 | 4.720 | 0.01128–0.01131 | 4.897 | 4.980 | 5.159 | 0.01171–0.01175 | 5.432 | 5.600 | 5.600 | 0.01220–0.01228 |
| 900 | | 4.489 | 4.609 | 4.768 | 0.01133–0.01135 | 4.903 | 4.994 | 5.299 | 0.01173–0.01177 | 5.533 | 5.700 | 5.790 | 0.01225–0.01230 |
| 1000 | | 4.500 | 4.630 | 4.869 | 0.01135–0.01137 | 4.959 | 5.048 | 5.299 | 0.01175–0.01179 | 5.633 | 5.750 | 5.800 | 0.01228–0.01236 |
| 1100 | | 4.510 | 4.631 | 4.791 | 0.01139–0.01142 | 4.936 | 5.099 | 5.306 | 0.01178–0.01181 | 5.637 | 5.770 | 5.810 | 0.01231–0.01241 |

**Table 4.** Bubble diameter (mm) as a function of water height for $T = 40\ °C$ and for gas volume flow rates $Q_1$–$Q_3$.

| Water Height (mm) | Water Temperature (°C) | Gas Volume Flow Rate $Q_1 = 1.5 \times 10^{-5}$ (m³/s) | | | | Gas Volume Flow Rate $Q_2 = 28 \times 10^{-5}$ (m³/s) | | | | Gas Volume Flow Rate $Q_3 = 50 \times 10^{-5}$ (m³/s) | | | |
|---|---|---|---|---|---|---|---|---|---|---|---|---|---|
| | | Lower | Middle | Upper | Range of Uncertainties | Lower | Middle | Upper | Range of Uncertainties | Lower | Middle | Upper | Range of Uncertainties |
| 100 | | 5.160 | 5.450 | 5.562 | 0.01204–0.01209 | 5.679 | 5.788 | 5.927 | 0.01239–0.01244 | 6.184 | 6.334 | 6.562 | 0.01279–0.01285 |
| 200 | | 5.160 | 5.450 | 5.562 | 0.01206–0.01210 | 5.679 | 5.788 | 5.927 | 0.01245–0.01248 | 6.184 | 6.334 | 6.562 | 0.01282–0.01288 |
| 300 | | 5.377 | 5.476 | 5.582 | 0.01206–0.01210 | 5.695 | 5.810 | 5.970 | 0.01247–0.01251 | 6.232 | 6.461 | 6.599 | 0.01286–0.01290 |
| 400 | | 5.391 | 5.476 | 5.635 | 0.01212–0.01216 | 5.742 | 5.820 | 5.980 | 0.01250–0.01253 | 6.256 | 6.536 | 6.800 | 0.01289–0.01295 |
| 500 | | 5.437 | 5.536 | 5.676 | 0.01214–0.01219 | 5.754 | 5.927 | 6.044 | 0.01255–0.01259 | 6.321 | 6.557 | 6.790 | 0.01294–0.01299 |
| 600 | T = 40 | 5.490 | 5.510 | 5.700 | 0.01218–0.01222 | 5.800 | 5.900 | 6.100 | 0.01258–0.01263 | 6.400 | 6.600 | 6.890 | 0.01298–0.01301 |
| 700 | | 5.510 | 5.540 | 5.670 | 0.01219–0.01225 | 5.900 | 5.990 | 6.090 | 0.01260–0.01266 | 6.380 | 6.800 | 6.950 | 0.01300–0.01309 |
| 800 | | 5.490 | 5.590 | 5.700 | 0.01221–0.01228 | 5.880 | 6.100 | 6.200 | 0.01266–0.01269 | 6.480 | 6.780 | 7.100 | 0.01308–0.01313 |
| 900 | | 5.560 | 5.650 | 5.800 | 0.01226–0.01232 | 5.990 | 6.200 | 6.400 | 0.01266–0.01270 | 6.500 | 6.850 | 7.170 | 0.01315–0.01320 |
| 1000 | | 5.550 | 5.723 | 5.900 | 0.01230–0.01235 | 6.002 | 6.400 | 6.600 | 0.01268–0.01274 | 6.700 | 6.900 | 7.250 | 0.01319–0.01325 |
| 1100 | | 5.590 | 5.690 | 5.990 | 0.01233–0.01238 | 6.200 | 6.450 | 6.700 | 0.01271–0.01276 | 6.900 | 7.050 | 7.340 | 0.01324–0.01335 |

**Table 5.** Bubble diameter (mm) as a function of water height for $T = 50\ °C$ and for gas volume flow rates $Q_1$–$Q_3$.

| Water Height (mm) | Water Temperature (°C) | Gas Volume Flow Rate $Q_1 = 1.5 \times 10^{-5}$ (m³/s) | | | | Gas Volume Flow Rate $Q_2 = 28 \times 10^{-5}$ (m³/s) | | | | Gas Volume Flow Rate $Q_3 = 50 \times 10^{-5}$ (m³/s) | | | |
|---|---|---|---|---|---|---|---|---|---|---|---|---|---|
| | | Lower | Middle | Upper | Range Of uncertainties | Lower | Middle | Upper | Range of Uncertainties | Lower | Middle | Upper | Range of Uncertainties |
| 100 | | 6.124 | 6.455 | 6.556 | 0.01310–0.01318 | 6.689 | 6.788 | 6.944 | 0.01350–0.01358 | 7.178 | 7.343 | 7.562 | 0.01401–0.01410 |
| 200 | | 6.355 | 6.476 | 6.578 | 0.01315–0.01324 | 6.695 | 6.817 | 6.970 | 0.01354–0.01364 | 7.243 | 7.476 | 7.687 | 0.01410–0.01419 |
| 300 | | 6.389 | 6.490 | 6.646 | 0.01319–0.01328 | 6.762 | 6.832 | 6.978 | 0.01360–0.01366 | 7.256 | 7.576 | 7.709 | 0.01418–0.01428 |
| 400 | | 6.464 | 6.521 | 6.686 | 0.01324–0.01329 | 6.775 | 6.927 | 7.034 | 0.01365–0.01368 | 7.332 | 7.587 | 7.698 | 0.01427–0.01446 |
| 500 | | 6.400 | 6.600 | 6.700 | 0.01327–0.01332 | 6.800 | 7.010 | 7.100 | 0.01366–0.01370 | 7.400 | 7.600 | 7.892 | 0.01450–0.01466 |
| 600 | T = 50 | 6.500 | 6.700 | 6.750 | 0.01330–0.01336 | 6.900 | 7.100 | 7.300 | 0.01370–0.01377 | 7.600 | 7.700 | 7.950 | 0.01466–0.01472 |
| 700 | | 6.550 | 6.650 | 6.800 | 0.01335–0.01339 | 6.850 | 7.050 | 7.350 | 0.01375–0.01380 | 7.550 | 7.900 | 8.019 | 0.01464–0.01471 |
| 800 | | 6.600 | 6.700 | 6.900 | 0.01336–0.01343 | 7.100 | 7.200 | 7.500 | 0.01379–0.01386 | 7.690 | 7.890 | 8.120 | 0.01475–0.01481 |
| 900 | | 6.700 | 6.800 | 7.020 | 0.01340–0.01347 | 7.200 | 7.400 | 7.700 | 0.01382–0.01389 | 7.800 | 8.010 | 8.130 | 0.01482–0.01487 |
| 1000 | | 6.800 | 6.900 | 7.100 | 0.01344–0.01349 | 7.290 | 7.500 | 7.800 | 0.01388–0.01395 | 7.900 | 8.200 | 8.340 | 0.01488–0.01490 |
| 1100 | | 6.750 | 6.890 | 7.090 | 0.01348–0.01354 | 7.250 | 7.480 | 7.780 | 0.01394–0.01400 | 7.910 | 8.150 | 8.314 | 0.01490–0.01500 |

The experiments for measuring bubble velocity were done in three regions of the bubble plume: the first region was over the injector region of the bubble generator, the second was in the middle region of the bubble plume, and the third was just under the free surface. The bubble velocity was defined by the equivalent bubble center using ellipsoidal approximations for the bubble shape. The equivalent bubble center was estimated from the vertical and horizontal lengths of each bubble in each image. The images were obtained by using JPEG image-converter software and Adobe After Effects CS6 image processing software after binarizing the images. The measurement uncertainty depends on the pixel resolution; for the bubble velocity, it was estimated to be 0.00508–0.01960 m/s, which is about 2% of the bubble velocity value, which reflects the accuracy of the experimental results. The calculation of bubble velocity, in this case, was reasonable and acceptable to find the effect of bubble velocity on the surface flow performance.

Figure 4 shows a sample of the bubble velocity as a function of water height in the tank for water temperatures of 30 °C. Where in this figure, the horizontal axis represents the water height (mm), the vertical axis represents the bubble velocity (m/s). The upper set of the three lines of the figure illustrates the relationship between the water height and the bubble velocity for $Q_3$, while the middle set is for $Q_2$ and the lower one is for $Q_1$. The green lines indicate the upper region (U) or upper level of the bubble plume inside the tank, while the red lines are for the middle level or region (M), and the blue lines are for the lower region or level (L). The lower level is just above the bubble generator (over the injector region of the bubble generator), the middle level is halfway between the bubble generator and the free surface, and the upper level is just under the free surface.

Tables 6–9 show the values of the bubble velocity as a function of water height in the tank for water temperatures of 20, 30, 40, and 50 °C, respectively. The data were collected for three gas volume flow rates ($Q_1 = 1.5 \times 10^{-5}$ m$^3$/s, $Q_2 = 28 \times 10^{-5}$ m$^3$/s, $Q_3 = 50 \times 10^{-5}$ m$^3$/s). The bubble velocity was calculated at three heights in the bubble plume: lower (L), middle (M), and upper (U). The first column of the tables shows the water height (mm), and the second column shows the water temperature, while the third set of columns (4 columns) elucidates the values of the bubble velocity for the three levels (lower, middle and upper regions of the tank) and for the gas volume flow rate $Q_1 = 1.5 \times 10^{-5}$ (m$^3$/s), while the fourth column of this set states the range of uncertainties. The fourth and the fifth sets are the same for gas volume flow rate $Q_2 = 28 \times 10^{-5}$ (m$^3$/s) and $Q_3 = 50 \times 10^{-5}$ (m$^3$/s), respectively. As in Figure 3, and Tables 2–5, the bubble velocity was calculated for the lower, middle, and upper levels in the bubble plume. These results show clearly that the bubble velocity increased with water height in the tank. Thus, the magnitude of bubble velocity increased with height in the bubble plume. Note also that the magnitude of the bubble velocity in the middle (upper) region was almost 1.5 times (twice) that in the lower region. These results are consistent with those of our earlier studies [67,68]. Furthermore, the bubble velocity increased as the water temperature increased. The results also show clearly that the bubble velocity increased by a factor of about 1.5 to 2 for each 10 °C increment in water temperature. As the temperature increased, viscosity and surface tension decreased, because the average speed of the molecules increased with temperature, which accelerated the liquid motion so that the viscosity decreased, which in turn allowed the bubble velocity to increase. Beyond that, the higher range of uncertainties occurred in the upper region, where the flow started to change its orientation from a horizontal into a vertical direction, forming the surface flow, and especially with the higher gas volume flow rates $Q_3 = 50 \times 10^{-5}$ (m$^3$/s), where the flow started to be turbulent.

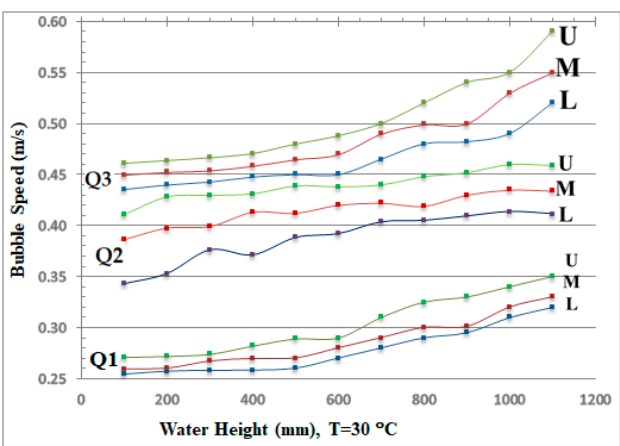

**Figure 4.** A sample of bubble speed as a function of water height for $T = 30\,°C$.

*3.3. The Void Fraction*

In multiphase flows, and particularly in the flows of two-phase gas–liquid, the void fraction is considered to be a highly valuable parameter. The void fraction is used to define and explain the interaction phase of dispersed bubbles. The relation between the experimental conditions and the action mechanism could be achieved using the analysis and prediction of the distribution of void fraction for the plume bubble. The result was applied in order to investigate the influence of the distribution of void fraction and the bubble plume structure in different situations. The two-dimensional measurement was performed. We used one camera to take the images in $x$, $y$ directions in this paper. The third dimension was not taken into account in the calculation of void fraction. The void fraction $\alpha$ was calculated directly by using $\alpha = Q/AV_b$ [48,51,56,57,60,67,69], where $A$ is the area of interest in the injector region (i.e., the injector surface of the bubble generator). The experiments that measure bubble velocity and the void fraction are of the same conditions. The bubble velocity $V_b \approx 0.08$ to $1.01$ m/s. The bubble rise velocity was unsteady at the beginning (at the nozzle exit) but reached the terminal upward velocity after a short time period. The measurement uncertainty for the bubble rise velocity was estimated to be $0.00508$–$0.01960$ m/s, which is about 2% of the value of bubble velocity. The relative velocity between the bubbles and the liquid flow corresponded well to the terminal upward velocity of a bubble in a quiescent liquid. The measurement uncertainty for the void fraction was estimated to be about 2% to 3% of the void fraction value. Therefore, in this stage, the two-dimensional measurement using our code helps grasp the internal flow structure. On the other hand, the calculations (of the bubble size, velocity, and void fraction) are not affected because the velocity is considered as the vertical component upward in the image processing. This kind of observation of the two-dimensional experiment for multiphase flows is a frequent subject treated in many kinds of research literature and reported by many authors and researchers [48,51,56,57,59,67,69–71]. The void fraction $\alpha$ was calculated at three heights in the bubble plume: lower (L), middle (M), and upper (U). The lower level was just above the bubble generator (over the injector region or surface of the bubble generator), while the middle level was halfway between the bubble generator and the free surface, and the upper level was just under the free surface. The data were collected for three gas volume flow rates ($Q_{g1} = 1.5 \times 10^{-5}$ m$^3$/s, $Q_{g2} = 28 \times 10^{-5}$ m$^3$/s, $Q_{g3} = 50 \times 10^{-5}$ m$^3$/s). A was calculated for each height separately, where for the lower level (L), A, the area of interest in the bubble plume, was the injector region (i.e., the injector surface of the bubble generator). This area was considered as the length and the width of the injection area of the bubble generator surface, while for the middle (M) and the upper (U) level, A was considered as the horizontal area of the bubble plume at each level, which was estimated as the dimensions of the two-phase zone containing bubbles in the longitudinal and transverse directions by image processing.

**Table 6.** Bubble speed (m/s) as a function of water height for $T = 20\ °C$ and for gas volume flow rates $Q_1$–$Q_3$.

| Water Height (mm) | Water Temperature (°C) | Gas Volume Flow Rate $Q_1 = 1.5 \times 10^{-5}$ (m³/s) | | | | Gas Volume Flow Rate $Q_2 = 28 \times 10^{-5}$ (m³/s) | | | | Gas Volume Flow Rate $Q_3 = 50 \times 10^{-5}$ (m³/s) | | | |
|---|---|---|---|---|---|---|---|---|---|---|---|---|---|
| | | Lower | Middle | Upper | Range of Uncertainties | Lower | Middle | Upper | Range of Uncertainties | Lower | Middle | Upper | Range of Uncertainties |
| 100 | | 0.090 | 0.138 | 0.170 | 0.00180–0.00340 | 0.205 | 0.229 | 0.239 | 0.00410–0.00478 | 0.253 | 0.257 | 0.271 | 0.00506–0.00542 |
| 200 | | 0.102 | 0.149 | 0.180 | 0.00204–0.00360 | 0.207 | 0.231 | 0.242 | 0.00414–0.00484 | 0.257 | 0.260 | 0.272 | 0.00514–0.00544 |
| 300 | | 0.105 | 0.150 | 0.180 | 0.00209–0.00360 | 0.212 | 0.236 | 0.247 | 0.00424–0.00494 | 0.258 | 0.267 | 0.274 | 0.00516–0.00548 |
| 400 | | 0.124 | 0.160 | 0.199 | 0.00248–0.00398 | 0.210 | 0.235 | 0.251 | 0.00420–0.00502 | 0.258 | 0.270 | 0.282 | 0.00516–0.00564 |
| 500 | | 0.122 | 0.161 | 0.200 | 0.00244–0.00400 | 0.228 | 0.239 | 0.255 | 0.00456–0.00510 | 0.261 | 0.274 | 0.290 | 0.00522–0.00580 |
| 600 | $T = 20$ | 0.132 | 0.174 | 0.210 | 0.00264–0.00420 | 0.234 | 0.250 | 0.260 | 0.00468–0.00520 | 0.269 | 0.279 | 0.300 | 0.00538–0.00600 |
| 700 | | 0.139 | 0.180 | 0.212 | 0.00278–0.00424 | 0.239 | 0.249 | 0.270 | 0.00478–0.00540 | 0.271 | 0.290 | 0.320 | 0.00542–0.00640 |
| 800 | | 0.145 | 0.182 | 0.230 | 0.00290–0.00460 | 0.248 | 0.255 | 0.280 | 0.00496–0.00560 | 0.289 | 0.295 | 0.330 | 0.00578–0.00660 |
| 900 | | 0.148 | 0.180 | 0.240 | 0.00296–0.00480 | 0.251 | 0.270 | 0.290 | 0.00502–0.00580 | 0.299 | 0.320 | 0.340 | 0.00598–0.00680 |
| 1000 | | 0.147 | 0.195 | 0.242 | 0.00294–0.00484 | 0.262 | 0.277 | 0.310 | 0.00524–0.00620 | 0.330 | 0.340 | 0.350 | 0.00660–0.00700 |
| 1100 | | 0.166 | 0.200 | 0.241 | 0.00332–0.00482 | 0.270 | 0.278 | 0.330 | 0.00540–0.00660 | 0.350 | 0.360 | 0.370 | 0.00700–0.00740 |

**Table 7.** Bubble speed (m/s) as a function of water height for $T = 30\ °C$ and for gas volume flow rates $Q_1$–$Q_3$.

| Water Height (mm) | Water Temperature (°C) | Gas Volume Flow Rate $Q_1 = 1.5 \times 10^{-5}$ (m³/s) | | | | Gas Volume Flow Rate $Q_2 = 28 \times 10^{-5}$ (m³/s) | | | | Gas Volume Flow Rate $Q_3 = 50 \times 10^{-5}$ (m³/s) | | | |
|---|---|---|---|---|---|---|---|---|---|---|---|---|---|
| | | Lower | Middle | Upper | Range of Uncertainties | Lower | Middle | Upper | Range of Uncertainties | Lower | Middle | Upper | Range of Uncertainties |
| 100 | | 0.254 | 0.259 | 0.271 | 0.00508–0.00519 | 0.343 | 0.386 | 0.411 | 0.00686–0.00772 | 0.436 | 0.450 | 0.461 | 0.00871–0.00900 |
| 200 | | 0.257 | 0.260 | 0.272 | 0.00514–0.00520 | 0.353 | 0.397 | 0.428 | 0.00706–0.00795 | 0.440 | 0.453 | 0.464 | 0.00880–0.00906 |
| 300 | | 0.258 | 0.267 | 0.274 | 0.00516–0.00534 | 0.376 | 0.399 | 0.430 | 0.00752–0.00798 | 0.443 | 0.454 | 0.467 | 0.00886–0.00908 |
| 400 | | 0.258 | 0.270 | 0.282 | 0.00516–0.00539 | 0.371 | 0.413 | 0.431 | 0.00742–0.00826 | 0.448 | 0.459 | 0.471 | 0.00895–0.00917 |
| 500 | | 0.260 | 0.270 | 0.289 | 0.00520–0.00540 | 0.388 | 0.412 | 0.439 | 0.00776–0.00824 | 0.460 | 0.460 | 0.490 | 0.00920–0.00920 |
| 600 | $T = 30$ | 0.270 | 0.280 | 0.290 | 0.00540–0.00560 | 0.392 | 0.429 | 0.438 | 0.00784–0.00858 | 0.450 | 0.470 | 0.488 | 0.00900–0.00940 |
| 700 | | 0.280 | 0.290 | 0.310 | 0.00560–0.00580 | 0.403 | 0.422 | 0.440 | 0.00807–0.00844 | 0.465 | 0.490 | 0.500 | 0.00930–0.00980 |
| 800 | | 0.290 | 0.300 | 0.325 | 0.00580–0.00600 | 0.405 | 0.419 | 0.450 | 0.00810–0.00838 | 0.480 | 0.499 | 0.520 | 0.00960–0.00998 |
| 900 | | 0.300 | 0.301 | 0.330 | 0.00600–0.00602 | 0.409 | 0.430 | 0.451 | 0.00818–0.00860 | 0.470 | 0.500 | 0.540 | 0.00940–0.01000 |
| 1000 | | 0.310 | 0.320 | 0.340 | 0.00620–0.00600 | 0.414 | 0.435 | 0.460 | 0.00827–0.00870 | 0.490 | 0.530 | 0.550 | 0.00980–0.01060 |
| 1100 | | 0.320 | 0.330 | 0.350 | 0.00640–0.00660 | 0.411 | 0.434 | 0.459 | 0.00823–0.00868 | 0.480 | 0.499 | 0.549 | 0.00960–0.00998 |

**Table 8.** Bubble speed (m/s) as a function of water height for $T = 40\ °C$ and for gas volume flow rates $Q_1$–$Q_3$.

| Water Height (mm) | Water Temperature (°C) | Gas Volume Flow Rate $Q_1 = 1.5 \times 10^{-5}$ (m³/s) | | | | Gas Volume Flow Rate $Q_2 = 28 \times 10^{-5}$ (m³/s) | | | | Gas Volume Flow Rate $Q_3 = 50 \times 10^{-5}$ (m³/s) | | | |
|---|---|---|---|---|---|---|---|---|---|---|---|---|---|
| | | Lower | Middle | Upper | Range of Uncertainties | Lower | Middle | Upper | Range of Uncertainties | Lower | Middle | Upper | Range of Uncertainties |
| 100 | | 0.416 | 0.473 | 0.488 | 0.00832–0.00947 | 0.498 | 0.518 | 0.555 | 0.00997–0.01037 | 0.597 | 0.650 | 0.737 | 0.01194–0.01299 |
| 200 | | 0.436 | 0.479 | 0.491 | 0.00873–0.00957 | 0.500 | 0.528 | 0.562 | 0.00999–0.01056 | 0.603 | 0.660 | 0.758 | 0.01207–0.01321 |
| 300 | | 0.457 | 0.480 | 0.493 | 0.00913–0.00960 | 0.507 | 0.539 | 0.573 | 0.01014–0.01078 | 0.615 | 0.691 | 0.773 | 0.01230–0.01383 |
| 400 | | 0.467 | 0.485 | 0.494 | 0.00934–0.00971 | 0.511 | 0.549 | 0.582 | 0.01022–0.01099 | 0.625 | 0.716 | 0.793 | 0.01250–0.01433 |
| 500 | | 0.450 | 0.480 | 0.520 | 0.00900–0.00960 | 0.525 | 0.570 | 0.589 | 0.01050–0.01140 | 0.630 | 0.719 | 0.800 | 0.01260–0.01438 |
| 600 | T = 40 | 0.459 | 0.488 | 0.515 | 0.00918–0.00976 | 0.550 | 0.574 | 0.590 | 0.01100–0.01148 | 0.660 | 0.720 | 0.835 | 0.01320–0.01440 |
| 700 | | 0.457 | 0.520 | 0.530 | 0.00914–0.01040 | 0.560 | 0.580 | 0.630 | 0.01120–0.01160 | 0.690 | 0.740 | 0.830 | 0.01380–0.01480 |
| 800 | | 0.468 | 0.510 | 0.540 | 0.00936–0.01020 | 0.590 | 0.596 | 0.660 | 0.01180–0.01192 | 0.700 | 0.760 | 0.890 | 0.01400–0.01520 |
| 900 | | 0.466 | 0.512 | 0.560 | 0.00932–0.01024 | 0.600 | 0.650 | 0.690 | 0.01200–0.01300 | 0.750 | 0.790 | 0.900 | 0.01500–0.01580 |
| 1000 | | 0.480 | 0.520 | 0.580 | 0.00960–0.01040 | 0.640 | 0.680 | 0.730 | 0.01280–0.01360 | 0.790 | 0.820 | 0.950 | 0.01580–0.01640 |
| 1100 | | 0.470 | 0.519 | 0.570 | 0.00940–0.01038 | 0.620 | 0.680 | 0.740 | 0.01240–0.01360 | 0.780 | 0.830 | 0.960 | 0.01560–0.01660 |

**Table 9.** Bubble speed (m/s) as a function of water height for $T = 50\ °C$ and for gas volume flow rates $Q_1$–$Q_3$.

| Water Height (mm) | Water Temperature (°C) | Gas Volume Flow Rate $Q_1 = 1.5 \times 10^{-5}$ (m³/s) | | | | Gas Volume Flow Rate $Q_2 = 28 \times 10^{-5}$ (m³/s) | | | | Gas Volume Flow Rate $Q_3 = 50 \times 10^{-5}$ (m³/s) | | | |
|---|---|---|---|---|---|---|---|---|---|---|---|---|---|
| | | Lower | Middle | Upper | Range of Uncertainties | Lower | Middle | Upper | Range of Uncertainties | Lower | Middle | Upper | Range of Uncertainties |
| 100 | | 0.522 | 0.556 | 0.597 | 0.01044–0.01111 | 0.635 | 0.687 | 0.724 | 0.01270–0.01375 | 0.764 | 0.807 | 0.847 | 0.01527–0.01615 |
| 200 | | 0.533 | 0.576 | 0.603 | 0.01066–0.01152 | 0.646 | 0.698 | 0.736 | 0.01292–0.01395 | 0.776 | 0.818 | 0.858 | 0.01552–0.01635 |
| 300 | | 0.544 | 0.587 | 0.615 | 0.01088–0.01175 | 0.659 | 0.707 | 0.747 | 0.01318–0.01415 | 0.787 | 0.828 | 0.877 | 0.01575–0.01655 |
| 400 | | 0.555 | 0.589 | 0.625 | 0.01110–0.01178 | 0.672 | 0.719 | 0.759 | 0.01343–0.01439 | 0.793 | 0.838 | 0.893 | 0.01586–0.01676 |
| 500 | | 0.560 | 0.590 | 0.630 | 0.01120–0.01180 | 0.670 | 0.720 | 0.760 | 0.01340–0.01440 | 0.780 | 0.820 | 0.880 | 0.01560–0.01640 |
| 600 | T = 50 | 0.559 | 0.600 | 0.650 | 0.01118–0.01200 | 0.680 | 0.730 | 0.770 | 0.01360–0.01460 | 0.820 | 0.860 | 0.920 | 0.01640–0.01720 |
| 700 | | 0.580 | 0.590 | 0.660 | 0.01160–0.01180 | 0.700 | 0.770 | 0.780 | 0.01400–0.01540 | 0.830 | 0.890 | 0.950 | 0.01660–0.01780 |
| 800 | | 0.587 | 0.600 | 0.670 | 0.01174–0.01200 | 0.750 | 0.772 | 0.800 | 0.01500–0.01544 | 0.880 | 0.900 | 0.960 | 0.01760–0.01800 |
| 900 | | 0.600 | 0.650 | 0.700 | 0.01200–0.01300 | 0.790 | 0.820 | 0.880 | 0.01580–0.01640 | 0.900 | 0.950 | 0.990 | 0.01800–0.01900 |
| 1000 | | 0.650 | 0.700 | 0.750 | 0.01300–0.01400 | 0.820 | 0.850 | 0.890 | 0.01640–0.01700 | 0.950 | 0.980 | 1.000 | 0.01900–0.01960 |
| 1100 | | 0.660 | 0.710 | 0.760 | 0.01320–0.01420 | 0.830 | 0.860 | 0.900 | 0.01660–0.01720 | 0.960 | 0.978 | 1.010 | 0.01920–0.01956 |

Figure 5 shows a sample of the void fraction $\alpha$ as a function of water height in the tank for water temperatures of 20 °C, where in this figure the horizontal axis represents the water height (mm), while the vertical axis represents the void fraction. The upper set of the three lines of the figure illustrates the relationship between the water height and the void fraction for $Q_3$, while the middle set is for $Q_2$ and the lower one is for $Q_1$. The green lines indicate the upper region (U) or upper level of the bubble plume inside the tank, while the red lines are for the middle level or region (M), and the blue lines are for the lower region or level (L). The lower level is just above the bubble generator (over the injector region of the bubble generator), the middle level is halfway between the bubble generator and the free surface, and the upper level is just under the free surface.

Tables 10–13 show the value of the void fraction $\alpha$ as a function of water height for a water temperature of 20, 30, 40, and 50 °C, respectively. The data were collected for three gas volume flow rates ($Q_1 = 1.5 \times 10^{-5}$ m$^3$/s, $Q_2 = 28 \times 10^{-5}$ m$^3$/s, $Q_3 = 50 \times 10^{-5}$ m$^3$/s). The void fraction was calculated at the three heights in the bubble plume: lower (L), middle (M), and upper (U). The first column of the tables shows the water height (mm), and the second column shows the water temperature, while the third set of columns (4 columns) elucidates the values of the void fraction for the three levels (lower, middle, and upper regions of the tank) for the gas volume flow rate $Q_1 = 1.5 \times 10^{-5}$ (m$^3$/s), while the fourth column of this set states the range of uncertainties. The fourth and the fifth sets are the same for gas volume flow rate $Q_2 = 28 \times 10^{-5}$ (m$^3$/s) and $Q_3 = 50 \times 10^{-5}$ (m$^3$/s), respectively. These results show clearly that the void fraction decreased as the water height increased. Furthermore, the void fraction decreased as the water temperature increased. In addition, the void fraction decreased by a factor up to about two for every 10 °C increments in temperature. Beyond that, the higher range of uncertainties occurred in the upper region where the flow started to change its orientation from a horizontal into a vertical direction, forming the surface flow, and especially with the higher gas volume flow rates $Q_3 = 50 \times 10^{-5}$ (m$^3$/s), where the flow started to be turbulent.

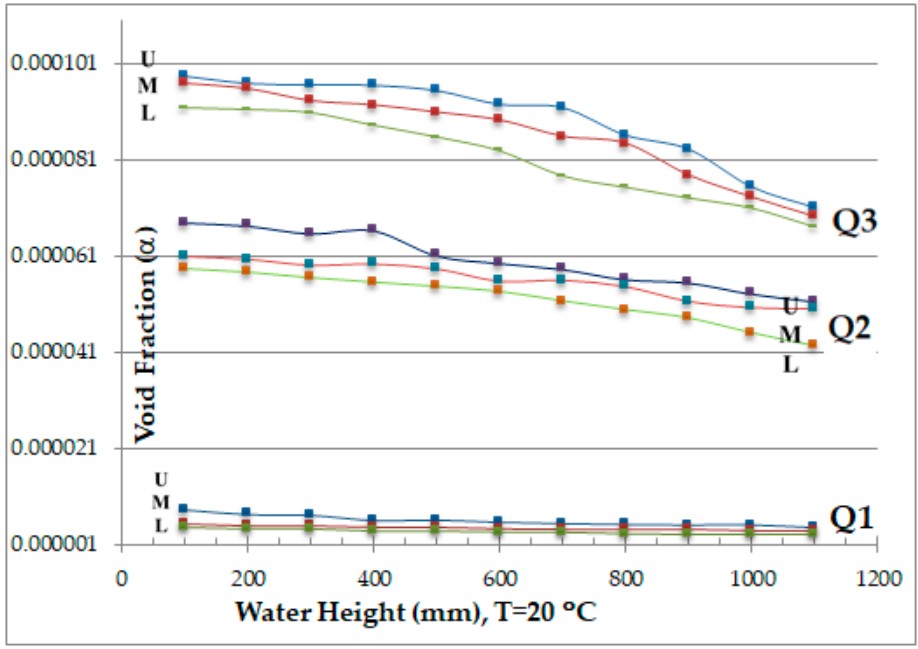

**Figure 5.** A sample of void fraction $\alpha$ as a function of water height for a water temperature of 20 °C and for gas volume flow rates $Q_1$, $Q_2$, and $Q_3$.

**Table 10.** Void fraction $\alpha$ as a function of water height for a water temperature of 20 °C and for gas volume flow rates $Q_1$–$Q_3$.

| Water Height (mm) | Water Temperature (°C) | Gas Volume Flow Rate $Q_1 = 1.5 \times 10^{-5}$ (m³/s) | | | | Gas Volume Flow Rate $Q_2 = 28 \times 10^{-5}$ (m³/s) | | | | Gas Volume Flow Rate $Q_3 = 50 \times 10^{-5}$ (m³/s) | | | |
|---|---|---|---|---|---|---|---|---|---|---|---|---|---|
| | | Lower | Middle | Upper | Range of Uncertainties | Lower | Middle | Upper | Range of Uncertainties | Lower | Middle | Upper | Range of Uncertainties |
| 100 | | 0.0000083 | 0.0000054 | 0.0000044 | 0.000000166–0.000000162 | 0.0000679 | 0.0000608 | 0.0000583 | 0.000001358–0.000001824 | 0.0000982 | 0.0000967 | 0.0000917 | 0.000001965–0.000002901 |
| 200 | | 0.0000073 | 0.0000050 | 0.0000041 | 0.000000146–0.000000150 | 0.0000673 | 0.0000602 | 0.0000575 | 0.000001345–0.000001806 | 0.0000967 | 0.0000956 | 0.0000914 | 0.000001935–0.000002869 |
| 300 | | 0.0000071 | 0.0000050 | 0.0000041 | 0.000000142–0.000000149 | 0.0000657 | 0.0000590 | 0.0000564 | 0.000001313–0.000001770 | 0.0000964 | 0.0000931 | 0.0000907 | 0.000001929–0.000002794 |
| 400 | | 0.0000060 | 0.0000047 | 0.0000037 | 0.000000120–0.000000140 | 0.0000663 | 0.0000592 | 0.0000555 | 0.000001326–0.000001777 | 0.0000964 | 0.0000922 | 0.0000882 | 0.000001927–0.000002767 |
| 500 | | 0.0000061 | 0.0000046 | 0.0000037 | 0.000000122–0.000000139 | 0.0000611 | 0.0000583 | 0.0000546 | 0.000001221–0.000001748 | 0.0000953 | 0.0000907 | 0.0000857 | 0.000001905–0.000002722 |
| 600 | T = 20 | 0.0000056 | 0.0000043 | 0.0000035 | 0.000000113–0.000000128 | 0.0000595 | 0.0000557 | 0.0000536 | 0.000001190–0.000001671 | 0.0000924 | 0.0000891 | 0.0000829 | 0.000001849–0.00002673 |
| 700 | | 0.0000054 | 0.0000041 | 0.0000035 | 0.000000107–0.000000124 | 0.0000583 | 0.0000559 | 0.0000516 | 0.00001165–0.000001677 | 0.0000917 | 0.0000857 | 0.0000777 | 0.000001835–0.00002572 |
| 800 | | 0.0000051 | 0.0000041 | 0.0000032 | 0.000000103–0.000000123 | 0.0000561 | 0.0000546 | 0.0000497 | 0.000001123–0.000001638 | 0.0000860 | 0.0000843 | 0.0000753 | 0.000001721–0.000002528 |
| 900 | | 0.0000050 | 0.0000041 | 0.0000031 | 0.000000101–0.000000124 | 0.0000555 | 0.0000516 | 0.0000480 | 0.000001109–0.000001547 | 0.0000832 | 0.0000777 | 0.0000731 | 0.000001663–0.000002331 |
| 1000 | | 0.0000051 | 0.0000038 | 0.0000031 | 0.000000101–0.000000115 | 0.0000531 | 0.0000503 | 0.0000449 | 0.000001063–0.000001508 | 0.0000753 | 0.0000731 | 0.0000710 | 0.000001507–0.000002194 |
| 1100 | | 0.0000045 | 0.0000037 | 0.0000031 | 0.000000090–0.000000112 | 0.0000516 | 0.0000501 | 0.0000422 | 0.000001031–0.000001502 | 0.0000710 | 0.0000691 | 0.0000672 | 0.000001421–0.000002072 |

**Table 11.** Void fraction $\alpha$ as a function of water height for a water temperature of 30 °C and for gas volume flow rates $Q_1$–$Q_3$.

| Water Height (mm) | Water Temperature (°C) | Gas Volume Flow Rate $Q_1 = 1.5 \times 10^{-5}$ (m³/s) | | | | Gas Volume Flow Rate $Q_2 = 28 \times 10^{-5}$ (m³/s) | | | | Gas Volume Flow Rate $Q_3 = 50 \times 10^{-5}$ (m³/s) | | | |
|---|---|---|---|---|---|---|---|---|---|---|---|---|---|
| | | Lower | Middle | Upper | Range of Uncertainties | Lower | Middle | Upper | Range of Uncertainties | Lower | Middle | Upper | Range of Uncertainties |
| 100 | | 0.0000029 | 0.0000029 | 0.0000027 | 0.000000059–0.000000086 | 0.0000406 | 0.0000361 | 0.0000325 | 0.000000812–0.000001082 | 0.0000571 | 0.0000553 | 0.0000539 | 0.000001142–0.000001658 |
| 200 | | 0.0000029 | 0.0000029 | 0.0000027 | 0.000000058–0.000000086 | 0.0000395 | 0.0000350 | 0.0000325 | 0.000000789–0.000001051 | 0.0000565 | 0.0000549 | 0.0000536 | 0.000001130–0.000001647 |
| 300 | | 0.0000029 | 0.0000028 | 0.0000027 | 0.000000058–0.000000084 | 0.0000370 | 0.0000349 | 0.0000324 | 0.000000741–0.000001047 | 0.0000561 | 0.0000547 | 0.0000533 | 0.000001123–0.000001642 |
| 400 | | 0.0000029 | 0.0000028 | 0.0000026 | 0.000000058–0.000000083 | 0.0000375 | 0.0000337 | 0.0000323 | 0.000000750–0.000001011 | 0.0000555 | 0.0000542 | 0.0000528 | 0.000001111–0.000001626 |
| 500 | | 0.0000029 | 0.0000028 | 0.0000026 | 0.000000057–0.000000083 | 0.0000359 | 0.0000338 | 0.0000317 | 0.000000717–0.000001014 | 0.0000541 | 0.0000541 | 0.0000507 | 0.000001081–0.000001622 |
| 600 | T = 30 | 0.0000028 | 0.0000027 | 0.0000026 | 0.000000055–0.000000080 | 0.0000355 | 0.0000325 | 0.0000318 | 0.000000711–0.000000974 | 0.0000553 | 0.0000529 | 0.0000509 | 0.000001105–0.000001587 |
| 700 | | 0.0000027 | 0.0000026 | 0.0000024 | 0.000000053–0.000000077 | 0.0000345 | 0.0000330 | 0.0000316 | 0.000000690–0.000000990 | 0.0000535 | 0.0000507 | 0.0000497 | 0.000001069–0.000001522 |
| 800 | | 0.0000026 | 0.0000025 | 0.0000023 | 0.000000051–0.000000075 | 0.0000344 | 0.0000332 | 0.0000309 | 0.000000688–0.000000997 | 0.0000518 | 0.0000498 | 0.0000478 | 0.000001036–0.000001495 |
| 900 | | 0.0000025 | 0.0000025 | 0.0000023 | 0.000000050–0.000000074 | 0.0000340 | 0.0000324 | 0.0000309 | 0.000000681–0.000000971 | 0.0000529 | 0.0000497 | 0.0000460 | 0.000001058–0.000001492 |
| 1000 | | 0.0000024 | 0.0000023 | 0.0000022 | 0.000000048–0.000000070 | 0.0000337 | 0.0000320 | 0.0000303 | 0.000000673–0.000000960 | 0.0000507 | 0.0000469 | 0.0000452 | 0.000001015–0.000001407 |
| 1100 | | 0.0000023 | 0.0000023 | 0.0000021 | 0.000000047–0.000000068 | 0.0000338 | 0.0000321 | 0.0000303 | 0.000000677–0.000000962 | 0.0000518 | 0.0000498 | 0.0000453 | 0.000001036–0.000001495 |

**Table 12.** Void fraction $\alpha$ as a function of water height for a water temperature of 40 °C and for gas **volume flow rates Q1–Q3**.

| Water Height (mm) | Water Temperature (°C) | Gas Volume Flow Rate $Q_1 = 1.5 \times 10^{-5}$ (m³/s) | | | | Gas Volume Flow Rate $Q_2 = 28 \times 10^{-5}$ (m³/s) | | | | Gas Volume Flow Rate $Q_3 = 50 \times 10^{-5}$ (m³/s) | | | |
|---|---|---|---|---|---|---|---|---|---|---|---|---|---|
| | | Lower | Middle | Upper | Range of Uncertainties | Lower | Middle | Upper | Range of Uncertainties | Lower | Middle | Upper | Range of Uncertainties |
| 100 | | 0.0000018 | 0.0000016 | 0.0000015 | 0.0000000358–0.0000000472 | 0.0000279 | 0.0000269 | 0.0000251 | 0.0000005588–0.0000008059 | 0.0000416 | 0.0000383 | 0.0000337 | 0.0000008328–0.0000011480 |
| 200 | | 0.0000017 | 0.0000016 | 0.0000015 | 0.0000000341–0.0000000467 | 0.0000279 | 0.0000264 | 0.0000248 | 0.0000005574–0.0000007910 | 0.0000412 | 0.0000377 | 0.0000328 | 0.0000008240–0.0000011295 |
| 300 | | 0.0000016 | 0.0000016 | 0.0000015 | 0.0000000326–0.0000000466 | 0.0000275 | 0.0000258 | 0.0000243 | 0.0000005491–0.0000007752 | 0.0000404 | 0.0000360 | 0.0000322 | 0.0000008086–0.0000010787 |
| 400 | | 0.0000016 | 0.0000015 | 0.0000015 | 0.0000000319–0.0000000461 | 0.0000273 | 0.0000253 | 0.0000239 | 0.0000005450–0.0000007604 | 0.0000398 | 0.0000347 | 0.0000314 | 0.0000007954–0.0000010414 |
| 500 | | 0.0000017 | 0.0000016 | 0.0000014 | 0.0000000331–0.0000000466 | 0.0000265 | 0.0000244 | 0.0000236 | 0.0000005304–0.0000007328 | 0.0000395 | 0.0000346 | 0.0000311 | 0.0000007893–0.0000010374 |
| 600 | T = 40 | 0.0000016 | 0.0000015 | 0.0000014 | 0.0000000325–0.0000000458 | 0.0000253 | 0.0000243 | 0.0000236 | 0.0000005063–0.0000007277 | 0.0000377 | 0.0000345 | 0.0000298 | 0.0000007534–0.000010360 |
| 700 | | 0.0000016 | 0.0000014 | 0.0000014 | 0.0000000326–0.0000000430 | 0.0000249 | 0.0000240 | 0.0000221 | 0.0000004973–0.0000007202 | 0.0000360 | 0.0000336 | 0.0000300 | 0.0000007207–0.0000010080 |
| 800 | | 0.0000016 | 0.0000015 | 0.0000014 | 0.0000000318–0.0000000438 | 0.0000236 | 0.0000234 | 0.0000211 | 0.0000004720–0.0000007008 | 0.0000355 | 0.0000327 | 0.0000279 | 0.0000007104–0.0000009814 |
| 900 | | 0.0000016 | 0.0000015 | 0.0000013 | 0.0000000320–0.0000000437 | 0.0000232 | 0.0000214 | 0.0000202 | 0.0000004641–0.0000006426 | 0.0000332 | 0.0000315 | 0.0000276 | 0.0000006630–0.0000009442 |
| 1000 | | 0.0000016 | 0.0000014 | 0.0000013 | 0.0000000310–0.0000000430 | 0.0000218 | 0.0000205 | 0.0000191 | 0.0000004351–0.0000006143 | 0.0000315 | 0.0000303 | 0.0000262 | 0.0000006294–0.0000009096 |
| 1100 | | 0.0000016 | 0.0000014 | 0.0000013 | 0.0000000317–0.0000000431 | 0.0000225 | 0.0000205 | 0.0000188 | 0.0000004491–0.0000006143 | 0.0000319 | 0.0000300 | 0.0000259 | 0.0000006375–0.0000008987 |

**Table 13.** Void fraction $\alpha$ as a function of water height for a water temperature of 50 °C and for gas volume flow rates $Q_1$–$Q_3$.

| Water Height (mm) | Water Temperature (°C) | Gas Volume Flow Rate $Q_1 = 1.5 \times 10^{-5}$ (m³/s) | | | | Gas Volume Flow Rate $Q_2 = 28 \times 10^{-5}$ (m³/s) | | | | Gas Volume Flow Rate $Q_3 = 50 \times 10^{-5}$ (m³/s) | | | |
|---|---|---|---|---|---|---|---|---|---|---|---|---|---|
| | | Lower | Middle | Upper | Range of Uncertainties | Lower | Middle | Upper | Range of Uncertainties | Lower | Middle | Upper | Range of Uncertainties |
| 100 | | 0.0000014 | 0.0000013 | 0.0000012 | 0.0000000358–0.0000000472 | 0.0000219 | 0.0000203 | 0.0000192 | 0.0000005588–0.0000008059 | 0.0000326 | 0.0000308 | 0.0000293 | 0.0000008328–0.0000011480 |
| 200 | | 0.0000014 | 0.0000013 | 0.0000012 | 0.0000000341–0.0000000467 | 0.0000216 | 0.0000200 | 0.0000189 | 0.0000005574–0.0000007910 | 0.0000320 | 0.0000304 | 0.0000290 | 0.0000008240–0.0000011296 |
| 300 | | 0.0000014 | 0.0000013 | 0.0000012 | 0.0000000326–0.0000000466 | 0.0000211 | 0.0000197 | 0.0000186 | 0.0000005491–0.0000007752 | 0.0000316 | 0.0000300 | 0.0000283 | 0.0000008086–0.0000010787 |
| 400 | | 0.0000013 | 0.0000013 | 0.0000012 | 0.0000000319–0.0000000461 | 0.0000207 | 0.0000194 | 0.0000183 | 0.0000005451–0.0000007604 | 0.0000314 | 0.0000297 | 0.0000278 | 0.0000007954–0.0000010414 |
| 500 | | 0.0000013 | 0.0000013 | 0.0000012 | 0.0000000331–0.0000000466 | 0.0000208 | 0.0000193 | 0.0000183 | 0.0000005304–0.0000007328 | 0.0000319 | 0.0000303 | 0.0000283 | 0.0000007893–0.0000010374 |
| 600 | T = 50 | 0.0000013 | 0.0000012 | 0.0000011 | 0.0000000325–0.000000045 | 0.0000205 | 0.0000191 | 0.0000181 | 0.000000506–0.0000007277 | 0.0000303 | 0.0000289 | 0.0000270 | 0.0000007534–0.000010360 |
| 700 | | 0.0000013 | 0.0000013 | 0.0000011 | 0.0000000326–0.0000000430 | 0.0000199 | 0.0000181 | 0.0000179 | 0.0000004973–0.0000007202 | 0.0000300 | 0.0000279 | 0.0000262 | 0.0000007207–0.0000010080 |
| 800 | | 0.0000013 | 0.0000012 | 0.0000011 | 0.0000000318–0.0000000438 | 0.0000186 | 0.0000180 | 0.0000174 | 0.0000004720–0.0000007008 | 0.0000283 | 0.0000276 | 0.0000259 | 0.0000007104–0.0000009814 |
| 900 | | 0.0000012 | 0.0000011 | 0.0000011 | 0.0000000320–0.0000000437 | 0.0000176 | 0.0000170 | 0.0000158 | 0.0000004641–0.0000006426 | 0.0000276 | 0.0000262 | 0.0000251 | 0.0000006630–0.0000009442 |
| 1000 | | 0.0000011 | 0.0000011 | 0.0000010 | 0.0000000310–0.0000000430 | 0.0000170 | 0.0000164 | 0.0000156 | 0.0000004351–0.0000006143 | 0.0000262 | 0.0000254 | 0.0000249 | 0.0000006294–0.0000009096 |
| 1100 | | 0.0000011 | 0.0000010 | 0.0000010 | 0.0000000317–0.0000000431 | 0.0000168 | 0.0000162 | 0.0000155 | 0.0000004491–0.0000006143 | 0.0000259 | 0.0000254 | 0.0000246 | 0.0000006375–0.0000008987 |

As one of the goals of this paper was to study how surface flow is impacted by different bubble parameters, the measurement of the length of the surface flow is of utmost importance. Table 14 shows the relationship between gas flow rate, water temperature, and the length of the bubble plume on the surface (the area that contains bubbles on the free surface is the "length of the surface flow" in the horizontal direction), which was measured from the video images, as shown in Figure 6. This dimension is the length of the surface flow parallel to the length of the take. In this case, the maximum length of the surface flow (376 mm) was small compared to the length of the tank (750 mm), and hence the plume was not affected by the sidewalls of the disproportionate aspect ratio of the tank. This dimension was calculated directly from the image processing by defining the length or distance on the surface that contained bubbles. The associated measurements' uncertainties depended on the pixel resolution of the images for the length of the surface flow of the image processing; it was estimated to be 0.5–1 mm, which reflects the accuracy of the experimental results, this estimation considered to be reasonable compared to the length of the surface flow.

**Table 14.** Water temperature and length of the bubble plume at the free surface for different gas flow rates.

| Gas Volume Flow Rate $Q$ (m³/s) | Water Temperature (°C) | Length of Surface Flow (mm) | Range of Uncertainties (mm) |
|---|---|---|---|
| $Q_1 = 1.5 \times 10^{-5}$ | 20 | 320 | 0.4960 |
| | 30 | 322 | 0.5120 |
| | 40 | 325 | 0.5265 |
| | 50 | 329 | 0.5593 |
| $Q_2 = 28 \times 10^{-5}$ | 20 | 345 | 0.6210 |
| | 30 | 348 | 0.6612 |
| | 40 | 350 | 0.7000 |
| | 50 | 353 | 0.7766 |
| $Q_3 = 50 \times 10^{-5}$ | 20 | 369 | 0.0849 |
| | 30 | 372 | 0.8928 |
| | 40 | 374 | 0.9350 |
| | 50 | 376 | 0.9776 |

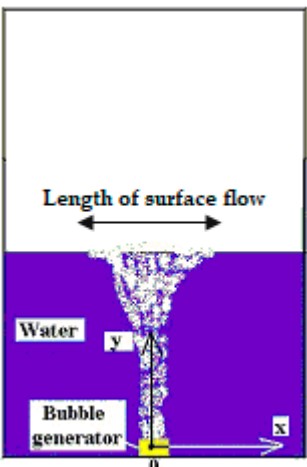

**Figure 6.** Length of surface flow.

The surface flow length was confirmed to increase as the gas flow rate increased. Moreover, this length increased as the water temperature increased. Thus, higher water temperature leads to stronger surface flow, all else being equal. The given results in Table 14 clearly show that the length of the surface flows increased by about 2 to 4 mm for each 10 °C increment in water temperature. Beyond that, the higher range of uncertainties occurred in this upper region, with the higher gas volume flow rate $Q_3 = 50 \times 10^{-5}$ (m$^3$/s), where the flow started to be turbulent, and the flow started to change its orientation from a horizontal into a vertical direction and forming the surface flow.

Given these results, the following properties would help to more efficiently generate surface flow: higher water temperature, larger bubble size, greater water height, and higher gas flow rate (and thus higher bubble velocity). The experiments covered different geometries, different regions (i.e., water height), and different bubble parameters (gas flow rate, water temperature, water height, bubble velocity, bubble size, and void fraction). These different conditions validate the gas-injection model.

## 4. Results Interpretation and Discussion

This section gives physical explanations of the underlying fluid mechanics and discusses the bubble generation mechanism to clarify how it depends on bubble size. The above results, including the flow visualization and image analysis of the bubble plume and bubble motion, indicate that the main upward liquid flow is driven along the bubble plume by the rising bubbles. The resulting flow is steady (especially when a small gas flow rate is given). The main flow then reaches the free surface, at which point the momentum of the upward flow is a maximum. Just under the free surface, the upward flow rapidly changes its orientation to horizontal flow. Note also that as the gas flow rate increases, the magnitude of velocity increases, and the effective area of the bubble plume (i.e., the length of the surface flow) expands horizontally. The liquid phase is continuously accelerated up to the free surface by the bubble motion in the vertical direction. The merit of using bubbles to generate a surface flow is indicated by the location of the liquid phase with the highest distortion rate, which is the region generating the surface flow (i.e., the vicinity of the free surface).

The actual bubble shape depends on the relative magnitudes of the forces acting on the bubble, such as surface tension and inertial forces [44]. For sufficiently small bubbles (e.g., an equivalent bubble diameter D < 1 mm in water), the surface-tension force predominates, and the bubble is approximately spherical. For bubbles of intermediate size, the effects of both surface tension and the inertia of the medium flowing around the bubble become important. As a result, intermediate-size bubbles exhibit very complex shapes and movements. These ellipsoidal bubbles often lack fore-and-aft symmetry and, in extreme circumstances, cannot be described by any simple regular geometry because of their significant shape fluctuations. Such complex shapes result from the superposition of various modes of fluctuations that have different amplitudes and frequencies. The dominant mode of shape fluctuations is quite periodic and is characterized by the extension and contraction of bubble height and width or vice versa, i.e., a variation in the bubble aspect ratio. Large bubbles (i.e., D > 18 mm in water) are generally dominated by inertial or buoyancy forces, with surface tension and viscosity of the liquid media being negligible. The bubble thus forms an approximate spherical cap. The fluctuations in the overall shape are suppressed; in other words, the main feature of the bubble-shaped fluctuation is the oscillation of the bubble base, especially in the edge region [44].

The bubble velocity and the surface flow velocity increase with the gas flow rate. Moreover, and as mentioned in our earlier papers [67,71,72], the largest kinetic energy is generated in the center of the bubble plume near the free surface. This observation confirms the hypothesis that the bubble plume generates a strong and wide surface flow over the bubble-generation system. Beyond that, the vorticity distribution in the area under the free surface is rapidly distorted, where the upward flow changes from vertical to horizontal. The liquid-phase flow changes from vertical to horizontal in the vicinity of the free surface.

Thus, the maximum velocity of the surface flow generated by the bubble plume occurs at the free surface. As a result, the surface flow is generated more efficiently by the bubble motion because the distortion point is near the free surface.

Three forces act on an air bubble rising in a column of water. The first force is the weight of the air in the bubble, which is exerted downwards. This force is constant and negligible compared with the other two forces. The second force is the buoyant force acting upward on the air bubble; the magnitude of this force equals that of the weight of the water displaced by the bubble. The initial net upward force resulting from these first two forces accelerates the bubble upward. As the bubble rises, the air pressure inside the bubble decreases because of the decrease in the height of the water column above the bubble. By the ideal gas law, the reduced pressure is accompanied by an increased bubble volume, which in turn increases the buoyant force. Thus, the bubble acceleration increases as the bubble rises, so its velocity increases at a faster rate. A viscous force (resistive force) also acts on the bubble to counter the upward motion of the bubble. The magnitude of the viscous force depends on the radius of the bubble, the water viscosity, and bubble velocity. The net result of these forces is a nonconstant bubble velocity.

Furthermore, note that we have ignored the inertial and viscous forces that are exerted during bubble formation and that usually occur at low flow rates. In this case, we should only consider surface tension and gravity.

To summarize, bubble formation may be analyzed by separating the problem into three regimes: static, dynamic, and turbulent. Dynamic bubble evolution is governed by inertial, viscous, surface tension, and buoyant forces. Controlling the bubble volume and bubble-formation frequency is essential in bubble dynamics. These types of motions and forces are described by the equations below.

Conservation of mass and momentum for an incompressible Newtonian fluid in the liquid and gaseous phases is embodied in the continuity equation,

$$\nabla \cdot \vec{v} = 0 \tag{1}$$

the momentum equation,

$$\rho \left( \frac{\partial v}{\partial t} + \vec{v} \cdot \nabla v \right) = -\nabla P + \rho_i g + \nabla \cdot [\mu_i \left( \nabla v + (\nabla v)^T \right)] \tag{2}$$

and the momentum equation with surface tension,

$$\rho \left( \frac{\partial v}{\partial t} + \vec{v} \cdot \nabla v \right) = -\nabla P + \rho_i g + \nabla \cdot [\mu_i \left( \nabla v + (\nabla v)^T \right)] + \sigma k \vec{n} \delta_s \tag{3}$$

The relative velocity between the bubbles and the flowing liquid corresponds well to the terminal upward velocity of the bubble in a quiescent liquid. In other words, the upward velocity corresponds well to the sum of the liquid velocity and the terminal slip velocity. This result indicates that the bubble motion itself is strongly governed by the balance between buoyant and drag forces near the free surface. The dynamic pressure-gradient force increased inertial force, and the other forces do not contribute significantly to the internal flow structure. This characteristic is one of the differences between this bubble scenario and a jet flow impinging on a solid wall, in which the dispersion is caused by a strong pressure-gradient force. If the flow field were similar to that of a jet impinging on a wall, the bubble velocity would vary because of the pressure gradient in the impinging region. However, in the case of bubbles, they do not experience a strong pressure-gradient force because atmospheric pressure is distributed evenly across the entire surface.

The velocity of the surface flow induced by the bubble plume in the vicinity of the free surface is greater than that in other regions. Moreover, the surface flow is generated particularly quickly in the vicinity of the free surface, which means that the increase in

the surface flow velocity corresponds to the increase in liquid volume flux pumped by the bubbles. Moreover, the surface flow velocity increases with the gas flow rate.

The bubble size is a very important factor because larger bubbles rise faster toward the free surface when there are fewer bubbles around them. This means that the period of momentum exchange between the gas and liquid phases is shortened. This reasoning is consistent with the numerical results reported in earlier papers [48,58,61]. For larger bubbles, the vortex size increases because the relaxation of the rapid distortion of liquid near the free surface occurs due to the large bubbles, which have a large relative upward velocity relative to the liquid motion.

If we compare the structure of the instantaneous flow for three different bubble sizes, we find that when the bubble injector is shifted upward, the upward liquid flow in the bubbling jet is not sufficiently developed near the free surface, so that the bubbling jet produces a laminar flow in all conditions. However, owing to the rapid distortion near the free surface, the surface flow involves a periodic fluctuation like a turbulent flow. Furthermore, large bubbles generate a wide surface flow area. The point at which the horizontal velocity has a local maximum depends on the bubble size. For large bubbles, the bubble plume forms and strongly induces the liquid flow. For small bubbles, the liquid flow is only induced in the water when the bubble plume becomes unstable. This instability is due to the local strong momentum exchange between bubbles and liquid.

The time evolution of the horizontal speed of the liquid phase, which is spatially averaged over the entire surface area, indicates that the larger bubbles more quickly generate surface flow, because larger bubbles have a greater velocity than smaller bubbles. In addition, smaller bubbles do not induce a high-speed surface flow because of the energy loss caused by vortex shedding. Large bubbles generate several large vortices or large-scale circulations of the liquid phase. Conversely, small bubbles generate small vortices near the free surface. Therefore, the speed of the surface flow is basically faster for large bubbles because of the low energy dissipation. However, for a shallow bubble injection point, the surface flow is reduced more than that for the small-bubble case.

In addition, the maximum velocity of the surface flow increases with the void fraction for small bubbles. The time evolution of the flow structure induced by the buoyant force for small bubbles indicates that the liquid is pumped upward and is diverted rapidly into a horizontal direction near the free surface. The width of the upward liquid flow remains narrow because of the upward acceleration, so that a strong distortion of the liquid phase occurs near the free surface. This leads to periodic vortex shedding.

These results led us to analyze the similarity between the laboratory model and an actual application. Moreover, the relationship between the bubble parameters (mean bubble diameter, bubble velocity, gas flow rate, void fraction, and the distance between the bubble generator and the free surface; i.e., the water height) was addressed to find the ideal parameters in a real application for the bubble plume to induce the strongest possible surface flow with the highest possible energy.

## 5. Conclusions

Bubble velocity and bubble volume are important for studying bubble motion because they are closely related to a void fraction. However, the volume of a bubble is difficult to calculate exactly, especially for distorted bubbles. The main reason surface flows induced by bubble plumes are used in so many fields, as mentioned above, is the simplicity of installation. The present paper is concerned with the characteristics of bubble parameters that induce surface flow, namely the gas volume flow rate, bubble size or mean bubble diameter, bubble velocity, void fraction, and internal two-phase flow structure of the bubble plume. This paper reports the results of an experimental investigation of bubble columns in which we studied the variation in bubble diameter and velocity as a function of water depth or the distance between the bubble generator and the free surface, which is equivalent to the water height in the tank (i.e., the height of the bubble plume). In addition, this paper evaluates how temperature affects the bubble parameters and bubble motion. Moreover,

the variation in the width of the bubble column was studied as a function of the gas flow rate. The results clearly show that the bubble plume, "which is a typical bubble flow," is a key phenomenon and is an effective tool for many applications. The study of bubble plumes is motivated by the possibility to improve its performance and thereby increase its range of applications in many engineering fields. The bubble plume was monitored by flow visualization, following which the images were analyzed, and the bubble motion was deduced and studied. The bubble parameters were then calculated. After visualizing the flow of different sections of bubble regions, the images were processed to clarify the relationship between the bubble parameters, the water height in the tank, and liquid temperature. The relationship between the bubble parameters was explained, enabling an improvement in the efficiency with which the surface flow is induced by the bubble plume (i.e., rapidly generating a strong, high, and wide surface flow over the bubble-generation system). Moreover, this approach allows us to control the parameters of the surface flow, such as thickness, length, and velocity, and thereby control the surface-flow performance. The maximum speed of the surface flow induced by a bubble plume is governed by the flow structure in the initial region, where the rising flow changes into a surface flow. The flow structure and bubble parameters are strongly affected by the gas flow rate, bubble size, and liquid temperature. The surface flow is thus expected to be an effective tool to support the function of an oil fence because it can generate a strong and wide surface flow over the bubble generation system, and it dampens wave motion. This will help for designing real systems that rely on surface flow generated by a bubble plume for controlling and collecting surface floating substances in marine systems, lakes, seas, rivers, and oceans (especially for the oil layer that forms after large oil spills). Surface flows generated by bubble plumes are thus crucial phenomena in various types of reactors, engineering processes, and industrial processes that involve a free surface. The main results can be summarized as follows:

(1) Bubble size increases as water temperature increases. The bubble diameter increases by about one mm for every 10 °C increment in water temperature. Moreover, the bubble size increases with increasing water height in the tank.

(2) Bubble velocity increases as water temperature increases. The results confirm that the bubble velocity increases by a factor of about 1.5 to 2 for every 10 °C increment in water temperature. In addition, the bubble velocity increases along the bubble plume as the height of water in the tank increases. Moreover, the magnitude of the bubble velocity at medium height in the tank is almost 50% greater than the magnitude of the bubble velocity just above the bubble generator. Furthermore, the magnitude of the bubble velocity just under the free surface is almost twice that just above the bubble generator.

(3) The void fraction decreases as the water height increases and as the water temperature increases. The void fraction decreases up to twofold for every 10 °C increment in water temperature.

(4) The length of the bubble plume on the surface (i.e., the area that contains bubbles on the free surface, or "the length of the surface flow" in the horizontal direction) increases with increasing water temperature. Thus, a higher water temperature leads to stronger surface flow. Moreover, this area increases approximately in proportion to the gas flow rate.

(5) Inside the bubble plume near the free surface, the bubble velocity and thus the velocity of the two-phase flow is greater, whereas the opposite is true in other regions. Thus, the generation of this high-speed flow is considered to be a primary contributor to the inducement of a strong surface flow.

(6) The higher range of uncertainties occurs in the upper region, where the flow starts to change its orientation from a horizontal into a vertical direction, forming the surface flow, and especially with the higher gas volume flow rates $Q_3 = 50 \times 10^{-5}$ (m$^3$/s), where the flow starts to be turbulent.

(7) The following properties are desired to efficiently generate surface flow: higher temperature, larger bubble size, deeper water, increased gas flow rate, and higher velocity.

**Funding:** This research received no external funding.

**Acknowledgments:** I thank Andrew Downer for his support in proofreading this paper.

**Conflicts of Interest:** The funders had no role in the design of the study; in the collection, analyses, or interpretation of data; in the writing of the manuscript, or in the decision to publish the results.

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
