# Peer review of "Improving the Performance of Surface Flow Generated by Bubble Plumes"

_fluids, doi:10.3390/fluids6080262_

Round 1

Reviewer 1 Report

This work is certainly relevant and the experimental technology appears sound, but there are many gaps that need to be addressed.

  1. The long list of references on lines 65, 70, and 73 is not a particularly helpful guide to the reader, as only a very small subset of these references is actually discussed in the text. At the sentence “A few publications (including those cited in the discussion above)…” could be where to discuss the publications that are more relevant to the work.
  2. More importantly, the author does not explain how the manuscript relates to past investigations. Is there any advancement in the image acquisition technique or in the way bubbles are identified? How does the range of conditions examined so far in the current literature compare with the experimental conditions here, etc. This work should not be left to the reader.
  3. Overall, the introduction is almost verbose in describing the importance and applications of bubble flow, without offering more details on the structure of the manuscript than the abstract does. More specifics would be helpful here, whereas the application-related comments (for instance on line 123 and 142-149) could be easily removed or moved to the Conclusion.
  4. Figures 3 to 18 are incongruous in the font size of the labels and under-resolved.
  5. Are Qg1, Qg2, Qg3 in the text the same as Q1, Q2, Q3?
  6. The author provides the minimum indispensable information about the image treatment procedure. Many parameters go into the process of sharpening, binarizing, smoothing the images, etc. What are they? What is the sensitivity of the results to those parameters?
  7. As two-dimensional pictures are acquired in the experiment, how is the third dimension taken into account in the calculation of the void fraction? If an assumption of axial symmetry is made, what was the author's approach to verify it? If the bubbles are not axis-symmetrical, it is possible that the uncertainty in evaluating their volume is much larger than the one assumed in the paper. How is this point discussed in the literature?
  8. Are there experiments that measured bubble velocity or void fraction at  some of the same conditions that the author uses, and how do they compare with these new results? The absence of a context to examine the new data is the main flaw which needs to be corrected before the manuscript is re-submitted.

Author Response

Dear Sirs,

Thank you very much for reviewing my paper and thank you for your comments.

Please note that the necessary changes were made to hopefully satisfy the requirements of the reviewers, where I have provided a track-change version of my revision as requested by the reviewer. I have considered all issues mentioned in the reviewers' comments carefully.

Please find below my response (in blue color) which is a list of changes or a rebuttal against each point that was being raised for the reviewers.

In the revised paper, please note that the corrections which were recommended by the reviewer were made in blue color.

I hope you consider my paper for publication in the Journal of Fluid

Thank you very much

Journal Fluids (ISSN 2311-5521)

Manuscript ID fluids-1243589

Type Article

Number of Pages 17

Title Improving the Performance of Surface Flow Generated by Bubble Plumes

Reviewer 1

Comments and Suggestions for Authors

This work is certainly relevant and the experimental technology appears sound, but there are many gaps that need to be addressed.

  1. The long list of references on lines 65, 70, and 73 is not a particularly helpful guide to the reader, as only a very small subset of these references is actually discussed in the text. At the sentence “A few publications (including those cited in the discussion above)…” could be where to discuss the publications that are more relevant to the work.

The long list of references on lines 65, 70, and 73  and other lines was corrected to be a particularly helpful guide to the reader. On the other hand, the introduction was rewritten in a better way to avoid any confusion to the readers.

A huge number of references was reduced to be easier for readers to understand the article. Moreover, this number was also broken into smaller blocks.

  1. More importantly, the author does not explain how the manuscript relates to past investigations. Is there any advancement in the image acquisition technique or in the way bubbles are identified? How does the range of conditions examined so far in the current literature compare with the experimental conditions here, etc. This work should not be left to the reader.

The introduction was rewritten in a better way to explain how the manuscript relates to past investigations. Moreover, the advancement in the image acquisition technique and in the way bubbles are identified was explained in more detail.

The range of conditions examined so far in the current literature compare with the experimental conditions here were added. Please find below more details on the explanations that were also added to the revised paper in (the introduction and in each part of calculating the bubble parameters).

The image processing used in this paper has advancement in the image acquisition technique of clarity and higher accuracy in identifying each bubble to find the bubble center, the bubble size, and the bubble velocity. our calculations use the time average of 250 consecutive frames from the image processing software (corresponding to 5 seconds of flow visualization). The average bubble diameter, bubble velocity, void fraction, and standard deviation are calculated by measuring over 20 000 bubbles in the local video images inside the bubble plume. These images were produced by taking local pictures of different regions. The experiments for measuring these parameters were done in three regions of the bubble plume: the first region was over the injector region of the bubble generator, the second was in the middle region of the bubble plume, and the third was just under the free surface. The literature mentioned in our introduction hardly shows an accurate technique similar to our technique in calculating these parameters. The visualized flows were recorded by a digital video camera (Panasonic HC-MDH2) at 50 frames per second. The digital images were preprocessed through the video-to-JPEG converter software and Adobe After Effects CS6 image processing software. The preprocessing entailed sharpening, binarizing, smoothing the images, and labeling bubbles. The code image processing software was used to enable identify each bubble after preprocessing the images.

The velocity of each bubble was calculated as the vertical velocity after identifying the position and the bubble center in each image and the position of the bubble center of the same bubble in the next consecutive image, hence, the distance between those two positions of the consecutive image was computed to be the movement of the bubble from one frame to another. By knowing the speed of the camera between two consecutive images (1/50), the velocity was calculated and then the average of all bubbles was computed for over than 20000 bubbles in the local video images inside the bubble plume for each level using image processing. The calculation of bubble velocity, in this case, can be reasonable and acceptable to find the effect of bubble velocity on the surface flow performance.

The bubble diameter was defined by the equivalent bubble diameter using ellipsoidal approximations for the bubble shape of each bubble and in each image. The equivalent bubble diameter was estimated from the vertical and horizontal lengths, which was obtained by using the image possessing software after binarizing the images. The minimum bubble size in the experiments was more than 2 mm. This kind of image processing for a bubbly two-phase flow is suitable for this kind of medium and large-scale flow system, it gives a reasonable accuracy especially when the number of bubbles reaches several thousand. Thus, averaged bubble diameter is necessary to carry out the present accuracy calculation. However, other techniques such as the two-fluid model and normal image processing which are the most popular local averaged models do not give high predictability owing to the lack of spatial resolution especially for the calculation of microbubble size. The microbubble cannot be measured with acceptable accuracy in that coded software in the two-phase region as it cannot be extracted from this image clearly due to the limitation in the pixel resolution. Hence, in our image acquisition technique the drawback of the code that it has a limitation of the calculation where it cannot calculate a small bubbles or microbubbles size that have a diameter of less than 1 mm. In our experiment cases, more than 98 % of the bubbles have a size of more than 2 mm. Although a few limited numbers of bubbles has a smaller size than 2 mm but those bubbles were eliminated to get higher accuracy.

  1. Overall, the introduction is almost verbose in describing the importance and applications of bubble flow, without offering more details on the structure of the manuscript than the abstract does. More specifics would be helpful here, whereas the application-related comments (for instance on line 123 and 142-149) could be easily removed or moved to the Conclusion.
  • The introduction was rewritten
  • The structure of this manuscript is explained and added to the introduction
  • The paragraph (The line 123 to 142-149) was removed and modified then moved to the Conclusion.

  1. Figures 3 to 18 are incongruous in the font size of the labels and under-resolved.

As the reviewer said that the figures are somehow confusing and need improvements and as there are many similar figures, samples of each result were illustrated in the revised paper including all the correction suggested by all reviewers, while other data were illustrated and listed as tables to avoid any confusion for the reader.  A detailed description in the body test, consistent color schemes, and symbols, and to be congruous in the font size of the labels and under-resolved all were considered.

  1. Are Qg1, Qg2, Qg3in the text the same as Q1, Q2, Q3?

All Qg were corrected and replaced to be Q

  1. The author provides the minimum indispensable information about the image treatment procedure. Many parameters go into the process of sharpening, binarizing, smoothing the images, etc. What are they? What is the sensitivity of the results to those parameters?

Yes, more information about the image treatment procedure was added. Also, the calculation of each bubble parameter was done in a separate section with more details to make it easier for the readers.

 The visualized flows were recorded by a digital video camera (Panasonic HC-MDH2) at 50 frames per second. The digital images were preprocessed by video-to-JPEG converter software and Adobe After Effects CS6 image processing software. The preprocessing entailed sharpening, binarizing, smoothing the images, and labeling bubbles. The code image processing software was used to enable identify each bubble after preprocessing the images. Where the images were converted to the computer to tread them by our image acquisition technique and software. In the image processing of our code, any noise or unknown objects in the images were first eliminated if available, then the images were smoothed to make them more clear and ready for bubble identification, after than the boundary of the bubble were identified and binarized, where the bubbles that have unclear boundary because of the light reflection or other reasons were sharpened then binarized. These popular kinds of preprocessing are necessary to many kinds of software or codes and usually do not affect the sensitivity of the results to the calculation of the parameters. This kind of preprocessing can be shown in many kinds of literature such as [18, 23, 27, 30, 32-35, 39, 41, 42, 44].

  1. As two-dimensional pictures are acquired in the experiment, how is the third dimension taken into account in the calculation of the void fraction? If an assumption of axial symmetry is made, what was the author's approach to verify it? If the bubbles are not axis-symmetrical, it is possible that the uncertainty in evaluating their volume is much larger than the one assumed in the paper. How is this point discussed in the literature?

In the taken images the bubble plume is located in the middle of the image and our observation showed that the bubbles have an almost two-dimensional motion in the x–y plane especially for low and medium gas flow rate except for a tiny degree of three-dimensional fluctuations due to turbulence of the flow with a high gas flow rate. The bubbles were considered to be two dimensions while indeed they were three-dimensional, but this consideration is accepted in or calculation as it does not affect the calculation and it was considered in the measurement uncertainty. The velocity of the flow is almost two-dimensional especially because the tank width is small comparing to other tank dimensions and there are no perpendicular components to the front and back walls in the flow field. Furthermore, the surface injection of the bubble generators is small comparing to the tank. The bubble generator was installed at the bottom center of the tank. The bubble generator was 300 mm long, 20 mm high, and 20 mm wide and made of transparent acrylic resin with 40 holes in the liner to avoid overlapping and/or coalescing bubbles and to assure accurate two-dimensional measurements. Each hole was 0.8 mm in diameter and separated by 7 mm from neighboring holes.

The two-dimensional measurement was performed. We used one camera to take the images in x, y directions in this paper. This third dimension was not taken into account in the calculation of void fraction as the void fraction a was calculated directly by using the equation a = Q/AVb [29, 30, 32, 33, 35, 36, 45, 47, 49,50, 51] where A is the area of interest in the injector region (i.e., the injector surface of the bubble generator). The bubble-rising velocity Vb ≈ 0.08 to 1.01 m/s.

The void fraction a was calculated at three heights in the bubble plume: lower (L), middle (M), and upper (U). The lower level was just above the bubble generator (over the injector region or surface of the bubble generator), while the middle level was about halfway between the bubble generator and the free surface, and the upper level was just under the free surface. The data were collected for three gas volume flow rates (Qg1 = 1.5 ´ 10−5 m3/s, Qg2 = 28 ´ 10−5 m3/s, Qg3 = 50 ´ 10−5 m3/s). A was calculated for each height separately. Where for the lower level (L), A: the area of interest in the bubble plume which was the injector region (i.e., the injector surface of the bubble generator). This area was considered as the length and the width of the injection area of the bubble generator surface. While for the middle (M), and the upper (U) level, A was considered as the horizontal area of the bubble plume at each level which was estimated as the dimensions of the two-phase zone containing bubbles in the longitudinal and transverse directions by image processing.

Let’s add these explanations to the revised paper. The measurement uncertainty for the bubble rise velocity was estimated to be about 2%. The relative velocity between the bubbles and the liquid flow corresponded well to the terminal upward velocity of a bubble in a quiescent liquid. The measurement uncertainty for the void fraction was estimated to be about 2% to 3% of the value of the void fraction (so uncertainty in evaluating their volume is not much larger than the one assumed in the paper but in the paper in was missing to mentioned that it is 2% to 3% of the value of the void fraction)

. Therefore in this stage, the two-dimensional measurement using our code helps grasp the internal flow structure. On the other hand, the calculations (of the bubble size, velocity, and void fraction) are not affected because the velocity is considered as the vertical component upward in the image processing. This kind of observation of the two-dimensional experiment for multiphase flows is a frequent subject treated in many research kinds of literature and reported by many authors and researchers: [29, 30, 32, 33, 35, 36, 45, 47, 49,50, 51],  (Abdulmouti, et. al. 2001, Hassan 2003, Murai and Matsumoto 1998 and 1999, Matsumoto and Murai 1995, Isao, et. al. 1993, Murai et. al. 1998, 2001, Hassan and Esam 2013, Leitch, et. al. 1989, Hara, et. al. 1984, Hussain and Narang 1984, Sun and Faeth 1986 a, b, Durst et. al. 1986, Hirt and Cook 1972, Murai et. al. 2001, Hussain and Narang 1984, Murai and Matsumoto 1998 and 1999, Matsumoto and Murai 1995, Hassan 2002, 2006, 2013, 2014, Hassan et. al. 2001, Milgram 1983, Fannelip, et. al. 1991, Hassan, et. al. 1992 a, b, among others).

Anyhow, more explanation on this issue was added to the revised paper.

  1. Are there experiments that measured bubble velocity or void fraction at  some of the same conditions that the author uses, and how do they compare with these new results? The absence of a context to examine the new data is the main flaw which needs to be corrected before the manuscript is re-submitted.

Yes, the experiments that measure bubble velocity and the void fraction are of the same conditions that were used, anyhow more details were added and the calculation of each bubble parameter was done in a separate section with more details to make it easier for the readers.

Thank you very much

Reviewer 2 Report

General comment:

This paper studies the effects of temperature on bubble parameters and bubble motion to better understand how they impact the formation of surface flow, with the ultimate goal of improving the efficiency of the generation of surface flow. Experiments were carried out to measure bubble parameters in a water column using image visualization technique. The author also performs an extensive literature review to explain the physics and complexity of bubble motions in liquid columns. Using the observations from the experimental data, the author explains how bubble size and velocity can impact the formation of surface flow, which can potentially be used in real world applications. Nevertheless, in the reviewer’s opinion, the paper can further be strengthened, particularly regarding the experimental methods and the discussion of the experimental results. The author is referred to the specific comments below for more information.

Specific comments:

  1. In Section 2, the author stated that the uncertainty of the measured void fraction was estimated to be 2-3% but did not provide details on how the uncertainty is calculated. The author is suggested to include the details on how it is determined. Furthermore, a table that summarizes the range of uncertainties of the measured parameters and bubble parameters is also recommended.
  2. The data presented in this work shows that the measured bubble parameters are parameters are influenced strongly by the water temperature. The author mentioned that a thermometer with a thermostat was used to monitor and control the water but did not provide information on where the thermometer was located in the tank. The author is suggested to include this detail. Furthermore, given the relatively large size of the experimental tank, how was the water temperature maintained at a relatively constant temperature throughout the entire tank?
  3. The author mentioned the equipment and software used for processing the images but did not include details on how the processing was done. The author is suggested to dedicate one or two paragraphs in Section 2 to describe the image-processing procedures and address the associated uncertainties, including the distortion of the observed bubble size due to refraction.
  4. In addition to the image-processing procedure, the author is suggested to include details on how bubble parameters such as the diameter and velocity were calculated by the software as well as the handling of the associated uncertainties. Was the bubble velocity calculated based on the difference in bubble positions in subsequent frames? If so, was the distance measured from the bubble interface or bubble center of mass?
  5. The author is suggested to include legends to Figures 3 to 14 or detailed descriptions in the body text to help readers understand the figures better. It is also recommended to use consistent color schemes and symbols for these figures. For instance, in Figure 4, the line for ‘Q3’ at location ‘U’ has a gray line with green markers.
  6. In Figures 11-14, the trends of the void fraction at the volume flow rate of Q1 are quite messy where local maxima and minima are observed. The author is suggested to provide an explanation or justification for the inconsistency.
  7. In Table 2, the width of the surface flow is shown to range from around 320 mm to 370 mm, which is greater than the width of the tank of 200 mm. Furthermore, that the tank was longer than it was wide. Did the author consider the possibility of the size of the plume being affected by the side walls and the disproportionate aspect ratio of the tank? If not, what are the justifications?
  8. Given that one of the goals of this paper is to study how surface flow is impacted by different bubble parameters, the measurement of the width of surface flow is of utmost importance. The author is suggested to provide more details regarding how the width is measured, including the criteria used for defining the width and the associated uncertainties.
  9. In Section 3, the author tends to present the observations from the figures and tables without providing much physical explanations. The author is recommended to provide more explanations and justifications for these observations to improve the overall quality of the discussion.

Author Response

Dear Sirs,

Thank you very much for reviewing my paper and thank you for your comments.

Please note that the necessary changes were made to hopefully satisfy the requirements of the reviewers, where I have provided a track-change version of my revision as requested by the reviewer. I have considered all issues mentioned in the reviewers' comments carefully.

Please find below my response (in blue color) which is a list of changes or a rebuttal against each point that was being raised for the reviewers.

In the revised paper, please note that the corrections which were recommended by the reviewer were made in blue color.

I hope you consider my paper for publication in the Journal of Fluid

Thank you very much

Journal Fluids (ISSN 2311-5521)

Manuscript ID fluids-1243589

Type Article

Number of Pages 17

Title Improving the Performance of Surface Flow Generated by Bubble Plumes

Reviewer 2

 General comment:

This paper studies the effects of temperature on bubble parameters and bubble motion to better understand how they impact the formation of surface flow, with the ultimate goal of improving the efficiency of the generation of surface flow. Experiments were carried out to measure bubble parameters in a water column using image visualization technique. The author also performs an extensive literature review to explain the physics and complexity of bubble motions in liquid columns. Using the observations from the experimental data, the author explains how bubble size and velocity can impact the formation of surface flow, which can potentially be used in real world applications. Nevertheless, in the reviewer’s opinion, the paper can further be strengthened, particularly regarding the experimental methods and the discussion of the experimental results. The author is referred to the specific comments below for more information.

Specific comments:

  1. In Section 2, the author stated that the uncertainty of the measured void fraction was estimated to be 2-3% but did not provide details on how the uncertainty is calculated. The author is suggested to include the details on how it is determined. Furthermore, a table that summarizes the range of uncertainties of the measured parameters and bubble parameters is also recommended.

The uncertainty of measurements can come from various sources such as the reference measurement device used to make the measurement, environmental conditions, the operator making the measurements, the procedure, the calculation of the image processing which is the most important factor, and many other sources. In our experiments, we made our best to minimize these factors. Especially that the experiments were repeated many times. The image processing used in this paper has advancement in the image acquisition technique of clarity and higher accuracy in identifying each bubble to find the bubble center, the bubble size, and the bubble velocity. our calculations use the time average of 250 consecutive frames from the image processing software (corresponding to 5 seconds of flow visualization). The average bubble diameter, bubble velocity, void fraction, and standard deviation are calculated by measuring over 20 000 bubbles in the local video images inside the bubble plume.

The “root sum of the squares” method and the stander deviations methods were applied to get higher accuracy (square the value of each uncertainty source). The combined standard uncertainty was calculated by squaring the value of each uncertainty component and add together all the results then calculate the square root of the result by finding the sum of squares. The result shows the combined standard uncertainty that we obtained. The dimensions of these measurements are as follows:

  • the distance of the bubble movements between one frame and another which affects the bubble velocity and void fraction.
  • The 2 dimensions of the area a for calculating the void fraction. A is the area of interest in the injector region which was considered as the horizontal area of the bubble plume at each level which was estimated as the dimensions of the two-phase zone containing bubbles in the longitudinal and transverse directions by image processing.
  • The 2 dimensions of the bubble size that affects the bubble diameters. The bubble diameter is defined by the equivalent bubble diameter using ellipsoidal approximations for the bubble shape of each bubble and in each image which was obtained by using the image possessing software after binarizing the images. The equivalent bubble diameter is estimated from the vertical and horizontal lengths of each bubble in each image.

The associated measurements' uncertainties depend on the pixel resolution of the images of the image processing for each dimension of the above calculation or measurements.

The measurement uncertainty for the bubble rise velocity was estimated to be about 2% of the bubble velocity value. The relative velocity between the bubbles and the liquid flow corresponded well to the terminal upward velocity of a bubble in a quiescent liquid. The measurement uncertainty for the void fraction was estimated to be about 2% to 3% of the void fraction value.  This ratio contains the 3 dimensions (distance of the bubble movements between one frame and another, 2 dimensions of the area A for calculating the void fraction).

While measuring the bubble size in our code, the bubbles were considered to be two dimensions while indeed they were three-dimensional, but this consideration is accepted in or calculation as it does not affect the calculation and it was considered in the measurement uncertainty. Another problem involved in this visualization approach was the mismatching of the refractive indices between bubbles and water and/or the impact of the inherently variant distance between the focus plane and the bubble, but this problem does not affect the calculations. The measurement uncertainty depends on the pixel resolution; for the bubble diameter, it was estimated to be 0.010 ~ 0.015 mm, which reflects the accuracy of the experimental results”.

The measurement of the length of the surface flow is calculated directly from the image processing by defining the length or the distance on the surface that contains bubbles. The associated measurements' uncertainties depend on the pixel resolution of the images of the image processing for the length of the surface flow, it is estimated to be 0.5 ~ 1mm, which reflects the accuracy of the experimental results this estimation considered to be reasonable comparing to the length f the surface flow.

A table that summarizes the range of uncertainties of the measured parameters and bubble parameters with more explanations were also added to the revised paper.

  1. The data presented in this work shows that the measured bubble parameters are influenced strongly by the water temperature. The author mentioned that a thermometer with a thermostat was used to monitor and control the water but did not provide information on where the thermometer was located in the tank. The author is suggested to include this detail. Furthermore, given the relatively large size of the experimental tank, how was the water temperature maintained at a relatively constant temperature throughout the entire tank?

Four heaters including thermometers with thermostats and sensors were used to read and control the water temperature, which was allowed to range from 20 to 50 °C to maintain the water temperatures. The heaters including thermometers with thermostats and sensors were located and distributed uniformly and equally in the right and lift sides of the upper and lower part of the tank.

The above explanation was added to the revised paper

  1. The author mentioned the equipment and software used for processing the images but did not include details on how the processing was done. The author is suggested to dedicate one or two paragraphs in Section 2 to describe the image-processing procedures and address the associated uncertainties, including the distortion of the observed bubble size due to refraction.

More details were added and the calculation of each bubble parameter was done in a separate section with more details to make it easier for the readers.

The image processing used in this paper has advancement in the image acquisition technique of clarity and higher accuracy in identifying each bubble to find the bubble center, the bubble size, and the bubble velocity. our calculations use the time average of 250 consecutive frames from the image processing software (corresponding to 5 seconds of flow visualization). The average bubble diameter, bubble velocity, void fraction, and standard deviation are calculated by measuring over 20 000 bubbles in the local video images inside the bubble plume. These images were produced by taking local pictures of different regions. The experiments for measuring these parameters were done in three regions of the bubble plume: the first region was over the injector region of the bubble generator, the second was in the middle region of the bubble plume, and the third was just under the free surface. The literature mentioned in our introduction hardly shows an accurate technique similar to our technique in calculating these parameters.

The visualized flows were recorded by a digital video camera (Panasonic HC-MDH2) at 50 frames per second. The digital images were preprocessed through the video-to-JPEG converter software and Adobe After Effects CS6 image processing software. The preprocessing entailed sharpening, binarizing, smoothing the images, and labeling bubbles. The code image processing software was used to enable identify each bubble after preprocessing the images.

The velocity of each bubble was calculated as the vertical velocity after identifying the position and the bubble center in each image and the position of the bubble center of the same bubble in the next consecutive image, hence, the distance between those two positions of the consecutive image was computed to be the movement of the bubble from one frame to another. By knowing the speed of the camera between two consecutive images (1/50), the velocity was calculated and then the average of all bubbles was computed for over 20000 bubbles in the local video images inside the bubble plume for each level using image processing. The calculation of bubble velocity, in this case, can be reasonable and acceptable to find the effect of bubble velocity on the surface flow performance.

The bubble diameter was defined by the equivalent bubble diameter using ellipsoidal approximations for the bubble shape of each bubble and in each image. The equivalent bubble diameter was estimated from the vertical and horizontal lengths, which was obtained by using the image possessing software after binarizing the images. The minimum bubble size in the experiments was more than 2 mm. This kind of image processing for a bubbly two-phase flow is suitable for this kind of medium and large-scale flow system, it gives a reasonable accuracy especially when the number of bubbles reaches several thousand. Thus, averaged bubble diameter is necessary to carry out the present accuracy calculation. However, other techniques such as the two-fluid model and normal image processing which are the most popular local averaged models do not give high predictability owing to the lack of spatial resolution especially for the calculation of microbubble size. The microbubble cannot be measured with acceptable accuracy in that coded software in the two-phase region as it cannot be extracted from this image clearly due to the limitation in the pixel resolution. Hence, in our image acquisition technique the drawback of the code that it has a limitation of the calculation where it cannot calculate a small bubbles or microbubbles size that have a diameter of less than 1 mm. In our experiment cases, more than 98 % of the bubbles have a size of more than 2 mm. Although a few limited numbers of bubbles has a smaller size than 2 mm but those bubbles were eliminated to get higher accuracy.

  1. In addition to the image-processing procedure, the author is suggested to include details on how bubble parameters such as the diameter and velocity were calculated by the software as well as the handling of the associated uncertainties. Was the bubble velocity calculated based on the difference in bubble positions in subsequent frames? If so, was the distance measured from the bubble interface or bubble center of mass?

The visualized flows were recorded by a digital video camera (Panasonic HC-MDH2) at 50 frames per second. The digital images were preprocessed through the video-to-JPEG converter software and Adobe After Effects CS6 image processing software. The preprocessing entailed sharpening, binarizing, smoothing the images, and labeling bubbles. The code image processing software was used to enable identify each bubble after preprocessing the images.

The velocity of each bubble was calculated as the vertical velocity after identifying the position of a bubble in each image and the position of the same bubble in the next consecutive image, hence, the distance between those two positions of the consecutive image was computed to be the movement of the bubble from one frame to another. By knowing the speed of the camera between two consecutive images (1/50), the velocity was calculated and then the average of all bubbles was computed for over 20000 bubbles in the local video images inside the bubble plume for each level using image processing. The bubbles which were not identified in the next consecutive image were eliminated from the calculations. Although the shooting speed was 50 frames per second, while the maximum measured bubble velocity was about 1 m/s. The velocity of a specific bubble was determined in this case as the calculation was conducted for each level of reign separately and the movement of the bubble in this small region was clearly captured. In fact the bubble motion is not vertical upward. It has the motion of zek-zak. But the velocity can be predicted in our calculation as the objective is to find the relationship between bubble velocity and other parameters.  The calculation of bubble velocity, in this case, can be reasonable and acceptable to find the effect of bubble velocity on the surface flow performance.

The bubble diameter was defined by the equivalent bubble diameter using ellipsoidal approximations for the bubble shape of each bubble and in each image. The equivalent bubble diameter was estimated from the vertical and horizontal lengths, which was obtained by using the image possessing software after binarizing the images. The minimum bubble size in the experiments was more than 2 mm.  This kind of image processing for a bubbly two-phase flow is suitable for this kind of medium and large scale flow system it gives a reasonable accuracy especially when the number of bubbles reaches several thousand. Thus, averaged bubble diameter is necessary to carry out the present accuracy calculation. However, the two-fluid model, which is one of the most popular local averaged models, does not give high predictability owing to the lack of spatial resolution especially for the calculation of microbubble size. The microbubble cannot be measured with acceptable accuracy in this code software in the two-phase region as it cannot be extracted from this image clearly due to the limitation in the pixel resolution. Hence, the drawback of the code that it has a limitation of the calculation where it cannot calculate a small bubbles or microbubbles size that have a diameter of less than 1 mm. In our experiment cases, more than 98 % of the bubbles have a size of more than 2 mm. Although a few limited numbers of bubbles has a smaller size than 2 mm but those bubbles were eliminated to get higher accuracy, also the overlapping bubbles (which were very limited) were eliminated from the calculations. One problem of measuring the bubble size in this code was that the bubbles were considered to be two dimensions while indeed they were three-dimensional, but this consideration is accepted in or calculation as it does not affect the calculation and it was considered in the measurement uncertainty. Another problem involved in this visualization approach was the mismatching of the refractive indices between bubbles and water and/or the impact of the inherently variant distance between the focus plane and the bubble, but this problem does not affect the calculations. The measurement uncertainty depends on the pixel resolution; for the bubble diameter, it was estimated to be 0.010 ~ 0.015 mm, which reflects the accuracy of the experimental results”.

  1. The author is suggested to include legends to Figures 3 to 14 or detailed descriptions in the body text to help readers understand the figures better. It is also recommended to use consistent color schemes and symbols for these figures. For instance, in Figure 4, the line for ‘Q3’ at location ‘U’ has a gray line with green markers.

As the reviewer said that the figures are somehow confusing and need improvements and as there are many similar figures, samples of each result were illustrated in the revised paper including all the correction suggested by all reviewers, while other data were illustrated and listed as tables to easily add the uncertainties to the calculation and to avoid any confusion for the reader.  A detailed description in the body test, consistent color schemes, and symbols, and to be congruous in the font size of the labels and under-resolved all were considered.

  1. In Figures 11-14, the trends of the void fraction at the volume flow rate of Q1 are quite messy where local maxima and minima are observed. The author is suggested to provide an explanation or justification for the inconsistency.

As the reviewer said that the figures are somehow confusing and need improvements and as there are many similar figures, samples of each result were illustrated in the revised paper including all the correction suggested by all reviewers, while other data were illustrated and listed as tables to avoid any confusion for the reader.  A detailed description in the body test, consistent color schemes, and symbols, and to be congruous in the font size of the labels and under-resolved all were considered. An explanation and justification for the inconsistency were provided in the revised paper.

  1. In Table 2, the width of the surface flow is shown to range from around 320 mm to 370 mm, which is greater than the width of the tank of 200 mm. Furthermore, that the tank was longer than it was wide. Did the author consider the possibility of the size of the plume being affected by the sidewalls and the disproportionate aspect ratio of the tank? If not, what are the justifications?

In fact, this dimension is the length of the surface flow parallel to the length of the take. So, to avoid any misunderstanding for the reader, let us change the word width into the word length and let us add a figure (Figure 15. Length of surface flow) to show this dimension. In this case, the maximum length of the surface flow (376 mm) is small compared to the length of the tank (750 mm) and hence the plume is not affected by the sidewalls of the disproportionate aspect ratio of the tank.

Let us add this explanation to the revised paper.

  1. Given that one of the goals of this paper is to study how surface flow is impacted by different bubble parameters, the measurement of the width of surface flow is of utmost importance. The author is suggested to provide more details regarding how the width is measured, including the criteria used for defining the width and the associated uncertainties.

As mentioned in the previous answer that, this dimension is the length of the surface flow parallel to the length of the take. So, to avoid any misunderstanding for the reader, let us change the word width into the word length and let us add a figure (Figure 15. Length of surface flow) to show this dimension. The measurement of the length of the surface flow is calculated directly from the image processing by defining the length or the distance on the surface that contains bubbles. The associated measurements' uncertainties depend on the pixel resolution of the images of the image processing for the length of the surface flow, it is estimated to be 0.5 ~ 1mm, which reflects the accuracy of the experimental results this estimation considered to be reasonable comparing to the length f the surface flow. A table that summarizes the range of uncertainties of calculating this dimension with more explanations were added to the revised paper

  1. In Section 3, the author tends to present the observations from the figures and tables without providing much physical explanations. The author is recommended to provide more explanations and justifications for these observations to improve the overall quality of the discussion.

Physical explanations were added to the results interpretation and discussions section. And more explanations and justifications for the observations were added after each parameter calculation to improve the overall quality of the discussion.

Thank you very much

Round 2

Reviewer 1 Report

The reviewer thanks the author for following through with the suggestions and for answering all the clarification question. The added part is a little less smooth and contains spell checking and syntax errors, however. Before publication, the author should carefully review, one more time, the manuscript.

Author Response

Answer: yes you are right, the manuscript still contains some spell checking and syntax errors. The manuscript was carefully reviewed by the author and by another speciesist. The necessary corrections were made.

Reviewer 2 Report

The authors have addressed the comments from the reviewers. However, there is an additional suggestion for the author. The addition of new tables in the manuscript helps readers to understand the results and the overall experimental work more clearly, however, in the reviewer's opinion, the presentation of these tables can be improved. For instance, in the PDF format, these tables are wider than the width of the page, thus some of the columns are missing. The authors are recommended to resize these tables or use smaller font size to make them fit the page better.  

Author Response

Answer: The presentations of the addition of new tables in the manuscript were improved. These tables were resized by using smaller font sizes to make them fit the page better and to be NOT wider than the width of the page, thus all missing columns became visible.